



# Large hemispheric difference in ultrafine aerosol concentrations in the lowermost stratosphere at mid and high latitudes

Christina J. Williamson[1,2], Agnieszka Kupc[2,3], Andrew Rollins[2], Jan Kazil[1,2], Karl D. Froyd[1,2], Eric A. Ray[1,2], Daniel M. Murphy[2], Gregory P. Schill[1,2], Jeff Peischl[1,2], Chelsea Thompson[1,2], Ilann Bourgeois[1,2], Thomas Ryerson[2*], Glenn S. Diskin[4], Joshua P. DiGangi[4], Donald R. Blake[5], Thao Paul V. Bui[6], Maximilian Dollner[3], Bernadett Weinzierl[3], Charles A. Brock[2]

[1]Cooperative Institute for Research in Environmental Sciences, University of Colorado, Boulder, CO 80309, U.S.A.
[2]Chemical Sciences Laboratory, National Oceanic and Atmospheric Administration, Boulder, CO 80305, U.S.A
[3]Faculty of Physics, Aerosol Physics and Environmental Physics, University of Vienna, 1090 Vienna, Austria
[4]NASA Langley Research Center, Hampton, VA 23681, USA
[5]Department of Chemistry, University of California Irvine, Irvine, CA 92697, U.S.A.
[6]Earth Science Division, NASA Ames Research Center, Moffett Field, California, U.S.A.
* Now at: Scientific Aviation, Boulder, CO 80301, U.S.A.

*Correspondence to*: Christina J. Williamson (christina.williamson@noaa.gov)

**Abstract.** The details of aerosol processes and size distributions in the stratosphere are important for both heterogeneous chemistry and aerosol-radiation interactions. Using in-situ, global-scale measurements of the size distribution of particles with diameters >3 nm from the NASA Atmospheric Tomography Mission (ATom), we identify a mode of ultrafine aerosol in the lowermost stratosphere (LMS) at mid and high latitudes. This mode is substantial only in the northern hemisphere (NH), and was observed in all four seasons. We also observe elevated $SO_2$, an important precursor for new particle formation (NPF) and growth, in the NH LMS. We use box modelling and thermodynamic calculations to show that NPF can occur in the LMS conditions observed on ATom. Aircraft emissions are shown as likely sources of this $SO_2$, as well as a potential source of ultrafine particles directly emitted by, or formed in the plume of the engines. These ultra-fine particles have the potential to grow to larger sizes, and to coagulate with larger aerosol, affecting heterogeneous chemistry and aerosol-radiation interactions. Understanding all sources and characteristics of stratospheric aerosol is important in the context of anthropogenic climate change as well as proposals for climate intervention via stratospheric sulphur injection. This analysis not only adds to the, currently sparse, observations of the global impact of aviation, but also introduces another aspect of climate influence, namely a size distribution shift of the background aerosol distribution in the LMS.

## 1 Introduction

Aerosols in the stratosphere have both radiative and chemical effects: scattering or absorbing light which cools or warms the Earth; and providing surfaces for heterogeneous chemical reactions. Aerosol lifetimes are much longer in the stratosphere than in the troposphere, increasing their influence (Crutzen, 2006). Recently, much attention has been given to climate intervention



by direct stratospheric aerosol injection, or by injection of gas-phase species that can form particles in the stratosphere (Shepherd, 2012;Council, 2015;Keith et al., 2014;MacMartin and Kravitz, 2019). For these reasons, it is imperative that the

background state, sources of and trends in stratospheric aerosols are well understood and can be accurately reproduced and predicted by global climate models. Currently, models struggle to reproduce observed aerosol size distributions in the lowermost stratosphere (LMS) (Solomon et al., 2011;Murphy et al., 2020).

Volcanic eruptions are a major source of stratospheric aerosols and precursor gases (Solomon et al., 2011;Vernier et al., 2011;Kremser et al., 2016). Aerosols and precursor gases can also enter the stratosphere from the tropical tropopause layer

(TTL) either quasi-isentropically into the extratropical LMS, or cross-isentropically into the tropical stratosphere. The latter can occur both via slow radiative heating or by overshooting convection (Kremser et al., 2016). The stratospheric aerosol background (in volcanically quiescent periods) is highly variable, and it is unclear how much this is affected by anthropogenic influence (Solomon et al., 2011). Increases in the amount of aerosol in the stratosphere have been observed recently, but whether this is due to anthropogenic emissions (Hofmann et al., 2009;Randel et al., 2010) or minor volcanic

eruptions (Vernier et al., 2011;Neely III et al., 2013;Brühl et al., 2015;Mills et al., 2016) remains unclear. Recent studies have shown pollution from the Asian summer monsoon affecting stratospheric aerosol (Yu et al., 2017;Lelieveld et al., 2018). These studies have focused on gas-phase species and larger particles. Pyro-convection is another possible source of particles in the LMS (Fromm and Servranckx, 2003;Damoah et al., 2006;Ditas et al., 2018;Yu et al., 2019), as is dust (Murphy et al., 2014). Two distinct aerosol accumulation modes (particles with diameters between 60 and 1000 nm) have

recently been identified in the LMS, one originating at higher altitudes in the stratosphere, and one entrained from the upper troposphere (UT) (Murphy et al., 2020).

Under a range of conditions, aerosols can form in-situ from the gas phase, in a process known as new particle formation. New particle formation (NPF) has been well documented in a variety of locations in the planetary boundary layer and free

troposphere (Kulmala et al., 2013;Williamson et al., 2019;Clarke et al., 2013). NPF in the TTL has been observed (Brock et al., 1995), as has wintertime NPF in the polar middle stratosphere (Campbell and Deshler, 2014;Wilson et al., 1992). Aircraft and, more recently, rocket emissions, are possible sources of small particles or precursor gases in the stratosphere (Lee et al., 2010;Schroder et al., 2000;Brock et al., 2000).

Atmospheric NPF is known to often involve sulphuric acid ($H_2SO_4$) and water (Kuang et al., 2008;Kulmala et al., 2013), highly oxidized organic molecules (Bianchi et al., 2019;Gordon et al., 2017;Riccobono et al., 2014;Stolzenburg et al., 2018a), and ions (Duplissy et al., 2016;Kirkby et al., 2011;Kirkby et al., 2016), where they are present. Ammonia and amines have been shown to be involved in NPF in the planetary boundary layer and in the free troposphere (Ball et al., 1999;Kürten et al., 2016;Bianchi et al., 2016;Smith et al., 2010;Almeida et al., 2013).






Most studies of atmospheric NPF are related to occurrence in the planetary boundary layer (PBL) (Kerminen et al., 2018). While it is obvious that the LMS is a very different environment from the PBL, it is worth drawing attention to the ways in which the unique environment of the LMS could be important for NPF. Firstly, the colder temperature of the LMS may influence NPF and growth in complex ways (Paasonen et al., 2013;Dada et al., 2017). Low temperatures found in the UT and

LMS reduce the vapour pressure of $H_2SO_4$, increasing rates of binary ($H_2SO_4 – H_2O$) nucleation (Easter and Peters, 1994). Low temperatures also decrease volatility, thereby increasing the semi-volatile organic species that can contribute to new particle formation and growth (Trostl et al., 2016;Stolzenburg et al., 2018b;Simon et al., 2020). Secondly, the drier environment of the LMS means that water is less available for NPF and growth than is usually the case in the PBL. Thirdly, total concentrations of aerosol and the related sinks for condensable vapours, clusters and small particles are generally higher

in the PBL, so PBL NPF is mostly observed where concentrations of precursor vapours are high, or under specific local conditions where condensation sinks are lower (Kerminen et al., 2018). In contrast, it may be possible for lower concentrations of condensable vapours to cause NPF in the LMS because of these low sinks. These low sinks in the LMS mean that we are essentially observing processes like NPF and growth in the LMS in slow motion when compared to the PBL. Low sinks, and low concentrations of more standard precursor gases mean that unconventional nucleation mechanisms may become important

in the LMS. It has been postulated that gas phase mercury could cause NPF under the right conditions, and one such possible event has been observed in the marine boundary layer near Antarctica (Humphries et al., 2015). Mercury containing aerosols have been observed in the lowermost stratosphere (Murphy et al., 2006). Iodine oxidation has been linked to atmospheric NPF in coastal regions (McFiggans et al., 2010;O'Dowd et al., 1999;O'Dowd et al., 2002) and over the arctic icepack (Baccarini et al., 2020). Iodine and bromine have both been observed in the UT (Volkamer et al., 2015;Dix et al., 2013) and stratosphere

(Koenig et al., 2020). Lastly, ozone levels are, apart from for some specific highly polluted areas in the PBL, higher in the LMS. This may well lead to different oxidation mechanisms than we typically consider in the troposphere.

It is imperative that we understand factors that regulate aerosol number in the lower stratosphere because this affects how condensed material in the stratosphere is apportioned to size, thus influencing heterogeneous chemistry, light scattering,

absorption, and sedimentation (Wilson et al., 2008). Ultrafine particles (3-12 nm in diameter) have the potential to influence all of these properties in the LMS. Climate intervention schemes that propose the injection of aerosols, or their precursor gases into the stratosphere, could be affected by the presence of ultrafine aerosol, which can remove gases and particles through condensation and coagulation.

Here we examine in-situ observations of ultrafine particles, and relevant gas-phase tracers and condensable species in the lowermost stratosphere in both hemispheres, to understand the prevalence, potential causes and importance of NPF in the LMS. Hemispheric differences in observed aerosol and cloud properties are a tool for understanding anthropogenic effects, since we can contrast the more anthropogenically influenced northern hemisphere (NH) with the less anthropogenically influenced southern hemisphere (SH). This technique has been previously used to constrain aerosol radiative forcing using





observations of cloud droplet number (McCoy et al., 2020). We use box-modelling, back-trajectories, thermodynamic calculations and emissions estimates to understand how NPF can occur in the LMS, factors influencing the amount of NPF, and other potential sources of ultrafine aerosol in this region.

## 2 Methods

We recently conducted global-scale in-situ aerosol observations on the NASA Atmospheric Tomography Mission (ATom)
(Wofsy et al., 2018). This mission consisted of four sets of near pole-to-pole flights on the NASA DC-8 over the remote Pacific and Atlantic oceans. Flight paths continuously scanned from ~0.2 to ~12 km altitude to measure the vertical structure of the atmosphere, and these paths were covered once in each of the four seasons to capture seasonal variability.

We measured aerosol size distributions from 3 nm to 4.5 µm using instruments inside the cabin of the DC-8 using Nucleation
Mode Aerosol Size Spectrometers (NMASS), modified Ultra-High Sensitivity Aerosol Spectrometers, and a Laser Aerosol Spectrometer (LAS, TSI Inc., St. Paul, MN, USA) (Williamson et al., 2018;Kupc et al., 2018;Brock et al., 2019). A second-generation Cloud, Aerosol and Precipitation Spectrometer (CAPS; Droplet Measurement Technologies), mounted under the aircraft's wing, extended the measured size range of aerosol and cloud size distributions covering the range between approximately 0.5 and 930 µm(Spanu et al., 2020). Size-resolved single particle composition measurements were made
using the Particle Analysis by Laser Mass Spectrometry instrument installed inside the cabin (Froyd et al., 2019). $SO_2$ observations sensitive at <100 parts per trillion by volume (pptv; nmol mol$^{-1}$) were made on the fourth set of flights (May 2018) using laser-induced fluorescence techniques (Rollins et al., 2017). Stratospheric air is identified using in-situ measurements of ozone and relative humidity. Ozone was measured using nitric oxide-induced chemiluminescence (Bourgeois et al., 2020), water vapour was measured by the Diode Laser Hygrometer (Diskin and Digangi, 2019;Diskin et
al., 2002), and global positioning and meteorological data were measured by the meteorological measurement system (Scott et al., 1990;Gaines et al., 1992;Chan et al., 1989). Trace gases were sampled using the Whole-Air-Sampler (WAS) system (Colman et al., 2001), and then analysed in the laboratory using multi-column gas chromatography utilizing flame ionization detectors (FIDs), electron capture detectors, and a mass selective detector (MSD). The CH3Cl is detected on the MSD and one of the FIDs while ethane is detected on another FID.


Measured aerosol size distributions are used to calculate condensation and coagulation rates. The coagulation kernel between two particles as a function of their diameters is calculated using the Fuchs expression for coagulation rate coefficient(Seinfeld and Pandis, 2006), at ambient pressure and temperature. We assume each particle to have the density of water (1000 kg m$^{-3}$). The condensation is calculated in the same way, substituting a molecule of sulphuric acid for one of the particles. The diameter
of a sulfuric acid molecule is calculated from bulk properties following the method from Lovejoy et al. (2004), neglecting





temperature effects on the probability distribution function of monomers, dimers and trimers. We sum the coagulation and condensation rate from all particles in the size distribution at each measurement time.

While ATom flights were not designed for stratospheric sampling, measurements were made of the LMS at mid-high latitudes
in both hemispheres on all deployments. Measurements were limited to altitudes below 13 km, so stratospheric air sampled was associated with a low tropopause and sometimes tropopause folds. For this reason, we choose to define the stratosphere here as ozone >250 parts per billion by volume (ppbv), altitude >8 km so as to be definitively above the tropopause, and relative humidity <10 % with respect to supersaturated water. For interhemispheric comparisons we mainly choose to examine stratospheric air with ozone <400 ppbv. Most of the southern hemisphere (SH) flights did not reach ozone >400 ppbv, whereas
higher ozone mixing ratios were sampled in the NH. Therefore, the ozone range from 250-400 ppbv was chosen to ensure consistent comparison between hemispheres. This stratospheric definition is consistent with that used in Murphy et al. (2020), which we will reference in this analysis. Figure 1 shows the flight paths of the ATom deployments, highlighting where the LMS was sampled.

Size distributions can be used to identify particles that have recently formed via NPF. Stable particles form at around 1.7 nm diameter from the growth of molecular clusters. Lifetimes of these newly formed particles are relatively short, on the oder of a few days (SM section 1), so their presence indicates recent NPF. The size distribution at the smallest sizes is measured by a battery of 5 (for ATom 1) or 10 (for ATom 2-4) condensation particle counters within the NMASS instruments that each measure the total number concentration of particles larger than a specified size. These concentrations are differenced to give
the total number of particles in 5-10 size bins from between 3 and 60 nm (Williamson et al., 2018). Recent NPF is diagnosed when number concentrations in the smallest measured size bin are larger than those in the next-smallest size bin by 3 times more than can be expected by uncertainty due to flow variation and counting statistics (Williamson et al., 2019).

Using a simple thermodynamic analysis, we assess whether LMS conditions prohibit or allow NPF via nucleation of the
negative ion binary $H_2SO_4$-$H_2O$ system. Bulk solutions have a characteristic $H_2SO_4$ saturation vapor pressure (SVP), that describes the thermodynamic driving force for condensation or evaporation of $H_2SO_4$. Similarly, each molecular cluster in the binary $H_2SO_4$/$H_2O$ system has a characteristic SVP value (Froyd and Lovejoy, 2012). SVP is a strong function of temperature and also depends on relative humidity (RH). $SVP_{max}$ is the maximum SVP value that growing clusters experience for a given RH. Using cluster thermodynamics that form the basis for the Model of Aerosols and Ions in the Atmosphere (MAIA) model
(Lovejoy et al., 2004;Kazil and Lovejoy, 2007;Kazil et al., 2007), we define the barrier to nucleation for the negative ion system as the ratio of $SVP_{max}$ to the typical daytime maximum $H_2SO_4$ concentration. When the ratio of $SVP_{max}$ to the partial pressure of $H_2SO_4$ ($p(H_2SO_4)$) is > 1, it is more energetically favourable for a cluster to evaporate $H_2SO_4$ than for molecules to condense onto that cluster. At ratios >10, NPF is highly improbable. When $SVP_{max}$/$p(H_2SO_4)$ <1, cluster growth is more energetically favourable than evaporation, and nucleation proceeds with no thermodynamic barrier.




To more quantitatively assess the effects of thermodynamics on NPF in the LMS, box modelling is performed using MAIA, following the methods detailed in Kupc et al. (2020). The model is run along back-trajectories, initiated at the aircraft location, which were calculated using the Traj3D trajectory model (Bowman, 1993) and the National Center for Environmental Prediction (NCEP) global forecast system (GFS) meteorology (2015). NCEP provides temperature, relative humidity, and

pressure along the trajectories for the MAIA runs. MAIA is initialized using condensation sinks and $SO_2$ concentrations estimated from ATom observations at similar latitudes and altitudes. The initial aerosol size distribution is specified as a lognormal mode with the given condensation sink. The geometric mean diameter (46 nm) and geometric standard deviation (2.8) were obtained by fitting a lognormal mode to the size distribution observed at the ATom measurement locations.

**3 More small particles and $SO_2$ are observed in the lowermost stratosphere in the Northern Hemisphere than in the**
**Southern Hemisphere**

Our observations show that the total number of aerosol particles in the NH LMS is higher than in the SH LMS in all seasons (Fig. 2a-d). The elevated numbers of stratospheric particles in the NH relative to the SH persists for all submicron sizes, but is largest at the smallest sizes (Fig. 2e). Analysis of number concentrations in the smallest observable size range shows statistically significant numbers of 3-7 nm particles relative to larger particles (see methods) (Williamson et al., 2019). This is

the case for substantial portions of the LMS in all four seasons (Fig. 3a, b) in the NH, but only for very small portions of the data in the SH. These 3-7 nm particles have lifetimes in the NH LMS of just a few days (see SM1). Particle number concentrations in the NH are between 4 and 100 times the SH concentrations in the LMS, and between 2 and 9 times the SH concentrations in the UT, and the number concentration of small particles are a larger fraction of the total aerosol number in the NH LMS than in the UT (Fig. 3c-f).


The majority of the aerosol surface area in the NH LMS is in the accumulation mode (60-1000 nm. While nucleation mode particles do not contribute substantially to the total aerosol surface area in the NH LMS, the Aitken mode (12-60 nm) can contribute around 10 % of the total in the NH (Fig. 4). Newly formed particles that grow to these sizes will influence heterogeneous chemistry in the stratosphere, with potential implications for ozone depletion (Hofmann and Solomon, 1989).

Aitken mode surface area is smaller in the SH LMS than in the NH LMS.

$SO_2$ oxidation is a primary source of atmospheric $H_2SO_4$ in the gas phase, which readily participates in new particle formation and growth (Kuang et al., 2008;Kulmala et al., 2013). Measurements of $SO_2$ mixing ratios with sensitivity to concentrations < 100 pptv were made on the fourth ATom deployment in May 2018. Mixing ratios of $SO_2$ in the NH LMS were several times

higher than in the SH (Fig 5a). Number concentrations of small particles show some correlation with $SO_2$, whereas number concentrations of larger particles show little to no correlation with $SO_2$ (Fig. 6). This observed correlation between $SO_2$





concentration and small particle concentrations suggests that the enhanced $SO_2$ in the NH LMS is likely to be a precursor vapour for this NPF. We hypothesize that nucleation is occurring in the LMS, and the amount of condensable vapor available is one of the factors controlling the number of small particles produced.

## 4 Observations and models suggest NPF occurs in the NH LMS

We now turn our attention to the possibility of in situ NPF within the LMS. Concentrations of nucleation mode particles in the NH LMS are sometimes correlated with elevated $SO_2$ in the NH LMS (Fig. 6), which is consistent with our understanding of $SO_2$ as a precursor for NPF. While it is possible that nucleation in the NH LMS involves gas phase species beyond sulfuric acid and water, the thermodynamics of binary nucleation in the sulfuric acid-water system is still relevant. If it is possible to nucleate aerosol from just sulfuric acid and water (with or without ions), then the addition of gas-phase organics or ammonia will only make it easier.

We have conducted thermodynamic modelling of NPF for the temperature, pressure and relative humidity observed in the NH LMS on ATom (see Methods). This modelling indicates, for $H_2SO_4$ concentrations of $10^6$ molecules cm$^{-3}$, whether it is energetically favourable for new particle formation via negative-ion-enhanced $H_2SO_4$-$H_2O$ nucleation to occur. Box modelling for $SO_2$ concentrations similar to those observed in the NH LMS (20 – 40 pptv) show this $H_2SO_4$ concentration is reasonable for the NH LMS, but is likely too high for the SH LMS (Fig. 7). Figure 8 shows $SVP_{max}/p(H_2SO_4)$ values calculated at the $RH_w$ (relative humidity with respected to supersaturated water) and temperature conditions sampled by the aircraft for a fixed $p(H_2SO_4)$ level of 0.1 pptv, or approximately $10^6$ molecules cm$^{-3}$ in the LMS (300 hPa, 220 K). The highest altitudes sampled during ATom favoured nucleation of the negative ion-mediated binary system, including many of the LMS segments (little to no thermodynamic barrier to nucleation under these conditions). Neutral binary nucleation is always less favourable than the negative ion system so calculated barriers for the negative ion system also denote barriers to the neutral system. This therefore indicates that NPF from sulfuric acid-water is possible in the NH LMS.

Where LMS $RH_w$ was below about 1-2 %, as the NH in October 2017 and SH in February 2017 (Fig. S1), large thermodynamic barriers prohibited nucleation. Fewer small particles were observed in the NH LMS in October than in other seasons (Fig. 2). No correlation is seen in the SH LMS between the occasional higher number concentrations of nucleation mode aerosol and $SO_2$ (Fig. 6). This lack of correlation, and the low concentrations of $SO_2$ in the SH LMS suggest that, if nucleation is sporadically occurring, or occurring at low rates, it is more likely to depend on species other than $H_2SO_4$.

Back-trajectories from observations in the NH LMS in May show 35 out of the 55 trajectories we ran spend days prior to observation in low relative humidity, stratospheric conditions (Fig 7a-c). 20 of the 55 trajectories experience more humid air, indicative of UT or tropopause conditions. The box model MAIA shows that 20 pptv of $SO_2$ (median observed $SO_2$ values in





the NH LMS in May were between 20 and 30 pptv, Fig. 5) produced $H_2SO_4$ on a diurnal cycle, with concentrations peaking

between 0.5 and 1.5 x $10^6$ cm$^{-3}$ (Fig. 7g). The sulfuric acid production consistently causes nucleation to occur in the box model. Nucleation produced noticeable increases in nucleation mode number concentrations when either starting $SO_2$ concentrations were at or above 40 pptv, or starting condensation sinks were at or below 1 x $10^{-4}$ s$^{-1}$ (Fig 7e-g). We note that 40 pptv $SO_2$ is higher than the median observed values in May in the NH LMS on ATom (Fig. 5), and observed condensation sinks were mostly above 1 x $10^{-4}$ s$^{-1}$ in this region (Fig. S2). In other seasons, the lower condensation sinks are closer to the modelled 1 x

$10^{-4}$ s$^{-1}$ where binary nucleation produced noticeable concentrations of small particles; however, we lack sensitive observations of $SO_2$ in these seasons. $SO_2$ concentrations in the model typically decrease by ~20% over 24 hours in these model runs, bringing mixing ratios fairly close to the observed concentrations within a few days.

The results from MAIA suggest that binary nucleation alone may be able to explain NPF in the NH LMS, although it is still

possible that other condensable species contribute. Condensable organic vapours are likely present at mixing ratios on the order of pptv in the NH LMS (Murphy et al., 2020), which could increase rates of particle nucleation in the LMS (Gordon et al., 2016;Kupc et al., 2020). Previous studies have shown the likely role of condensable organics in the growth of newly formed particles in tropical UT (Kupc et al., 2020), and it is conceivable the low observed concentrations of organics in the NH LMS play a similar role. The case for involvement of condensable vapours beyond $H_2SO_4$ must not be overstated however, as there

are a number of uncertainties involved in modelling this phenomenon, including a lack of high sensitivity $SO_2$ measurements in three of the seasons.

Chamber studies have shown that the rate at which aerosols nucleate in the presence of sulfuric acid, water and ammonia mixing ratios > ~0.2 pptv increases with the amount of ammonia at temperatures relevant to the LMS (208-223K). At higher

temperatures (248K) this was also shown for even lower ammonia mixing ratios (Kürten et al., 2016). Single particle composition measurements of larger particles show only slight neutralization of sulphate in the NH LMS, suggesting gas-phase mixing ratios of ammonia <1 pptv (Fig. S3). Larger concentrations of ammonia have been observed in the UT (Höpfner et al., 2016), but so far only in outflow from the ASM which we did not detect in the ATom observations. Therefore, we cannot exclude that ammonia may play a role in the NH LMS new particle formation observed on ATom. Even where the SH and NH

$SO_2$ concentrations are similar, nucleation mode aerosol concentrations tend to be higher in the NH than in the SH (Fig 6a). This suggests that species other than $H_2SO_4$, may also be more abundant in the NH LMS than the SH, and play a role in nucleation and/or growth of ultrafine aerosol. Similarly, we note that nucleation mechanisms that may be uncommon in the boundary layer, such as those including halogens or mercury may be playing a role here, but do not have the observations to test these ideas.


NPF is a highly non-linear process with respect to precursor concentrations. The regions containing LMS air measured on ATom were heterogeneous mixes of more stratosphere-like and more troposphere-like air. The increased production of small

particles in the higher RH$_w$ cases in MAIA (Fig. 7 f-g) means that we cannot rule out that NPF is preferentially taking place in more troposphere-like air in these regions, followed by transport/mixing on short timescales. While the bulk UT has been ruled out as a source of the NH LMS small particles in the arguments given above, and we would expect NPF occurring in more troposphere-like air to lead to higher concentrations of small particles at lower potential temperatures, contrary to the observations shown in figure 9, observations of aerosol size distributions, chemical composition and gas-phase precursors of NPF further above the tropopause are needed to completely rule out the influence of the UT on NPF in the NH LMS.

With regard to SH LMS observations of nucleation mode aerosol, while the concentrations were consistently lower than those observed in the NH LMS, concentrations between 125 and 175 std. cm$^{-3}$ were regularly observed in the SH LMS in all seasons (Fig. 2). These may also indicate NPF, albeit at a slower rate than in the NH. The slower rate is implied not only by the lower concentrations, but also by the longer lifetimes of nucleation mode particles in the LMS in the SH compared to the NH (Supplemental Materials S2), because of the lower concentrations of particles of all sizes observed here. Observations in the SH LMS did not extend to high enough altitudes to enable us to draw conclusions from the vertical structure.

**5 The observed ultrafine particles and SO₂ are not being transported into the NH LMS, but rather being either formed or directly emitted into this region**

The main potential transport routes for particles and SO$_2$ into the NH LMS are transport from the tropics, either within the stratosphere from the tropical lower stratosphere, or quasi-horizontally across the tropopause from the tropical UT, vertical entrainment from the NH UT, downwelling from deeper in the stratosphere, and quasi-horizontal transport from the polar vortex in winter.

New particles have previously been shown to form in the tropical lower stratosphere and be transported northward in the stratosphere (Brock et al., 1995), and also to form in the tropical UT (Williamson et al., 2019;Clarke et al., 2013;Clarke and Kapustin, 2002) from where they could be transported quasi-horizontally across the tropopause into the NH LMS. Both of these transport pathways into the NH LMS occur on timescales of weeks to months or longer, which is inconsistent with the transport of newly formed particles, with lifetimes of a few days (Supplemental Materials S2, Fig. S4. S5, S6), from the tropics or NH UT into the NH LMS. Although quasi-isentropic transport on timescales less than a week has recently been observed in relation to extratropical cyclones and small-scale mixing (Kunkel et al., 2019), these small-scale phenomena are not frequent enough to produce the consistent elevated number concentrations of small particles we observed in the NH LMS.

Transport from the tropics would lead to increased particle and SO$_2$ concentrations in the LMS in both hemispheres, potentially with some enhancement in the summer hemisphere due to the location of the ITCZ (Vellinga and Wood, 2002;Chiang and Bitz, 2005;Broccoli et al., 2006). Since we only observed large numbers of small particles in the LMS in the NH in all seasons,



and SO$_2$ observations in NH spring also showed much higher concentrations in the NH than the SH LMS, transport from the tropics cannot explain the observations. ATom observations in the tropics were not at high enough altitude to reach similar potential temperatures to those observed in the LMS at higher latitudes, but SO$_2$ observations were taken at these potential temperatures in the tropics on the NASA POSIDON mission in October 2016 (Rollins et al., 2018). SO$_2$ mixing ratios in the NH LMS are larger than those measured at similar potential temperatures in the tropical UT (Fig. 5c), which also makes it

unlikely that the increased NH LMS SO$_2$ was transported quasi-isentropically from tropics.

Evidence suggests that entrainment from the NH UT at middle and high latitudes cannot explain the observed concentration and spatial distribution of SO$_2$ in the NH LMS. The distribution of SO$_2$ with potential temperature in the NH LMS (Fig. 5) shows the largest mixing ratios around 340 K, with lower mixing ratios at lower potential temperature (correlating with lower

altitude). This profile strongly argues against a tropospheric source, and instead suggests direct emission of SO$_2$ within the LMS.

Nucleation mode aerosol have been observed deep into the LMS (O$_3$ > 800 ppbv), and not just close to the tropopause (Fig. 3). Because the lifetime of these particles is ~days, the enhanced abundance of these particles distant from the tropopause

suggests that they have either formed in the LMS or been directly emitted in the LMS, rather than having been transported from the UT. In addition, the vertical profile of nucleation mode particles and SO$_2$ in the NH LMS are very different to the vertical profile of larger particles of tropospheric origin (Murphy et al., 2020), and other tracers of UT mixing, such as ethane and chloroform (both have tropospheric sources and lifetimes on the order of weeks to months (Parrish et al., 1992;Khalil et al., 1983), and particles from biomass burning (Fig. 10). This supports the argument that the small particles and SO$_2$ in the NH

LMS are not a result of vertical transport from the midlatitude UT nor horizontal transport from the tropics.

The vertical distributions of newly formed particles and SO$_2$ are also evidence against downwards transport of SO$_2$ from carbonyl sulphide (OCS) oxidation deeper in the stratosphere (Crutzen, 1976;Chin and Davis, 1995;Sheng et al., 2015;Brühl et al., 2012;Rollins et al., 2017). O$_3$ concentrations increase in the stratosphere with distance from the tropopause, therefore

O$_3$ is an effective tracer of depth of stratospheric air. If SO$_2$ and the resulting nucleation mode particles were originating from OCS oxidation deep in the stratosphere, we would expect them to be correlated with O$_3$. We observe no such correlation in the NH LMS (Fig. 7), and therefore conclude that the NH excess SO$_2$ is not from OCS oxidation. Some correlation can be seen in the SH LMS between SO$_2$ and O$_3$, suggesting OCS oxidation may be contributing to these lower, background levels of SO$_2$.

O$_3$ concentrations increase in the stratosphere with distance from the tropopause, therefore O$_3$ is an effective tracer of depth of stratospheric air. If the observed SO$_2$ originated from OCS oxidation deep in the stratosphere, we would expect it to be correlated with O$_3$. We observe no such correlation in the NH LMS (Fig. 9), and therefore conclude that the NH excess SO$_2$ is not from OCS oxidation. Some correlation can be seen in the SH LMS between SO$_2$ and O$_3$, suggesting OCS oxidation may





be contributing to these lower, background levels of $SO_2$. NPF can take place in descending air in the polar vortex in winter,
and quasi-horizontal transport is a potential pathway for these particles to then enter the LMS. However, similar to SO2, no
correlation between nucleation mode particle concentrations and O3 is observed (Fig. 9), suggesting they do not originate
deeper in the stratosphere. Furthermore, this pathway occurs only in winter, which does not explain the observed seasonal
persistence of high nucleation mode aerosol in the NH LMS, and should occur in both hemispheres, which is inconsistent with
the observed hemispheric difference in LMS nucleation mode aerosol concentrations (Fig. 2, 3).

**6. Aircraft are the most likely source of small particles in the NH LMS, either by direct emission of particles or emission
of precursor vapours for NPF**

Air traffic is largely concentrated in the NH (Lee et al., 2010) and most emissions occur at ~10 km altitude (Schröder et al.,
2000). It has previously been observed that small particles are directly produced from aircraft (Brock et al., 2000;Kinsey et al.,
2010), either as particles or forming from the gas phase immediately upon exiting the engine. Aircraft also directly emit $SO_2$
into the atmosphere. Here we examine aircraft as a potential source of the observed elevated concentrations of nucleation mode
aerosol, both through direct emissions, and through NPF resulting $SO_2$ oxidation.

We use the Community Emissions Data System (CEDS) 2014 emissions database (Hoesly et al., 2018) to look at the global
distribution and amount of $SO_2$ emitted by aircraft in 2014 (Fig. 11). CEDS calculates aircraft emissions based on Lee et al.
(2010) and Lamarque et al. (2010), which use $SO_2$ emissions indices between 0.6 and 1 g $kg^{-1}$ fuel burned, with an average of
0.6 g $kg^{-1}$. Global aircraft movements are taken from the AERO2K database (Eyers et al., 2004) with altitudes parameterized
from statistical analysis of air traffic data. Fuel consumption was calculated using the PIANO aircraft performance model and
then scaled up to the International Energy Agency statistics of kerosene sales data to account for documented underestimates.
We take the tropopause height from MERRA2 reanalysis temperatures in 2014.


Emission indices of particles with diameters between 3 and 10 nm (equivalent to our measured nucleation mode, 3-12 nm) in
aircraft exhaust while cruising at 10.7 km have been measured as ~$10^{16}$ -$10^{17}$ particles per kg of aircraft fuel for low sulphur
fuel (fuel sulphur content = 2.6 ppmm) (Brock et al., 2000). We calculate that 19 % of all aircraft $SO_2$ emissions occurred in
the NH between the tropopause and 13 km (this upper level is chosen to align with the maximum altitude of the ATom flights
in order to relate the emissions directly to the observed regions), and 0.05 % in the SH for the same region (Fig. 8), and assume
that small particle emissions have a very similar spatial distribution to $SO_2$ emissions. Global aircraft fuel usage rate in 2018
was 327.1 Tg/yr (Lee et al., 2020), which, assuming the same spatial distribution in 2018 as in 2014, gives maximum total
number of nucleation mode particles emitted from aircraft in the NH and SH LMS 626 x $10^{25}$ and 1.74 x $10^{25}$ respectively.



While particles lifetimes in non-plume LMS conditions are expected to be longer in the SH than the NH, for aircraft emissions, it is a reasonable first approximation to use a lifetime of 2 days for nucleation mode particles in the LMS in both hemispheres (Supplemental Materials S2). Calculating the volume of the region defined above as the LMS to be $3.12 \times 10^{17}$ m$^3$ and $3.65 \times 10^{17}$ m$^3$ for the NH and SH respectively, we get ambient concentrations of nucleation particle emitted by aircraft in the LMS of 110 cm$^{-3}$ and 0.26 cm$^{-3}$ in the NH and SH respectively. Taking an average temperature and pressure of 220 K and 300 hPa

for the LMS regions, and converting to standard temperature and pressure (STP), this gives maximum concentrations of 300 std. cm$^{-3}$ and 0.71 std. cm$^{-3}$ for the NH and SH respectively. Minimum aircraft emissions of nucleation mode aerosol from Brock et al. (2000) are a factor of 10 lower than the maximum we have used here, leading to concentrations of 30 and 0.07 std. cm$^{-3}$ for the NH and SH respectively.

The observed median nucleation mode aerosol concentrations in the NH LMS on ATom were between 50 and 600 std. cm$^{-3}$ (Fig. 9), a similar order of magnitude to concentrations of nucleation mode particles calculated from aircraft emissions. Median observed nucleation mode particle concentrations in the SH LMS were between 2 and 10 std. cm$^{-3}$ (Fig. 9). Since observed concentrations in the SH LMS are larger than concentrations calculated from aircraft emissions, this could imply a small additional source of nucleation mode aerosol, although a substantial uncertainty exists from the more varied lifetime in

the SH between plume and non-plume conditions. Direct aircraft emissions are of a similar order of magnitude to the NH LMS enhancement observed on ATom. While within the large uncertainties of this comparison, the nucleation mode concentrations calculated from aircraft emissions are lower than the observed concentrations. This therefore does not discount the possibility of NPF occurring outside of aircraft plumes in the NH LMS as a result of elevated SO$_2$ concentrations.


Figure 11a compares the spatial distribution of aircraft SO$_2$ emissions in the NH LMS with the spatial distribution of nucleation mode number concentrations measured in the same region on ATom emissions. No obvious correlation between nucleation mode number concentrations and flight corridors is evident, with some of the highest nucleation mode number concentrations observed in the LMS around 1000 km away from flight corridors. Median zonal windspeeds encountered in the LMS on ATom

were between 10 and 30 m/s, with the 75$^{th}$ percentile of these windspeed between ~20 and 35 m/s (Fig. S7), therefore 1000 km is around half a day's transport from the flight corridor. Even with this fast, zonal transport, we might expect some correlation of number concentration with flight corridor locations, but airborne observations with denser sampling to compare highly trafficked with less trafficked regions in the NH, and times around commercial flight times, would be needed to give a clearer picture of this. We hope that data from the In-Service Aircraft for a Global Observing System (IAGOS) program, which

includes an instrument package measuring concentrations of particles with diameters >13 nm, will provide useful information from commercial aircraft when the data are made publicly available (Bundke et al., 2015).



From the global distributions of aircraft $SO_2$ emissions from the CEDS database for 2014, we calculate the resulting $SO_2$ concentration in the LMS, assuming a 1-month lifetime of $SO_2$ in this region (see SM section 2). Fast zonal mixing in the
LMS, as evidenced above, is assumed in these calculations. An in-depth study of aircraft emissions shows a 23% increase in $SO_2$ emissions between 2014 and 2018 (Lee et al., 2020), therefore we multiply the calculated concentrations of $SO_2$ from 2014 by a factor 1.23 to get the expected 2018 concentrations and compare these to ATom observations in the LMS as a function of potential temperature (Fig. 11d). Concentration of observed $SO_2$ in the NH LMS are very similar to the expected aircraft emissions, and have a very similar vertical structure with a maximum occurring around 340 K in both emissions and
observations. Emitted and observed $SO_2$ concentrations in the SH LMS are very similar, although observed concentrations are consistently a few pptv higher. While this is certainly within the uncertainty of these calculations, higher observed concentrations compared with aircraft emissions in the SH LMS is also likely to indicate that while aviation emissions dominate in the NH where we see large amounts of air-traffic, other sources of $SO_2$ in the LMS exist, and become apparent in the SH LMS where there is currently very little air-traffic. Summing the 2014 zonal average $SO_2$ flux, and multiplying by
surface area of each grid box and a factor 1.23 to update for 2018 air-traffic, we get a total flux of 0.279 Tg yr$^{-1}$, with totals between the tropopause and 13 km in the NH and SH respectively of 0.066 and 0.000183 Tg yr$^{-1}$.

Nonetheless, CEDS shows aircraft emissions capable of producing $SO_2$ concentration in the LMS of similar orders of magnitude to those observed in ATom in both hemispheres, and with the same striking hemispheric difference, and the same
vertical structure in the NH. This strongly supports the argument that the observed elevated $SO_2$ concentrations in the NH LMS are caused by emissions from aircraft. The consistency between measured $SO_2$ mixing ratios and values calculated to be produced by aircraft emissions, together with the MAIA simulations along air parcel trajectories, implies that ion-assisted, binary homogeneous NPF in the NH LMS is a likely consequence of aircraft emissions.

In addition, we note that non-methane volatile organic compounds (NMVOCs) emitted from aircraft follow a similar distribution to $SO_2$ from aircraft (Fig. S8). Even through the emitted flux of NMVOCs from aircraft is of a similar magnitude to $SO_2$, we expect the steady-state mixing ratio of NMVOCs would be lower than that of $SO_2$ because they likely have shorter lifetimes in LMS conditions (Balkanski et al., 1993;Tsigaridis et al., 2014). The co-emission of NMVOCs and $SO_2$ from aircraft means aircraft are a potential source of two of the most likely species involved in aerosol nucleation and growth in the
LMS, though we cannot at this stage quantify the fraction of condensable organic vapours observed by Murphy et al. (2020) in the NH LMS that can be accounted for by aircraft emissions.

**7. Other potential direct sources of nucleation mode particles and NPF precursors in the NH LMS**





Other candidates for direct emissions of nucleation mode particles and $SO_2$ in the NH LMS are pyro-convection, the Asian Summer Monsoon (ASM), and volcanoes. Pyroconvection and the ASM are unlikely to contribute substantially for the
following reasons:

1) There was no significant correlation between small particles in the LMS and biomass burning particles (Fig. 10), which suggests that pyro-convection is not the major source of the small particles we observed. In October, a slight anticorrelation can be seen between the vertical structure of mass of particles from biomass burning and the number of nucleation mode
aerosols, suggesting that influence from fires may suppress NPF in the LMS, or that the additional surface area from biomass burning particles shortens the lifetime of newly formed particles.
2) While the ASM has been shown to be a source of $SO_2$ and particulate matter in the NH LMS, this is true only during the monsoon season from June to September (Yu et al., 2017). Since $SO_2$ lifetime in the stratosphere is ~1 month (SM Section S2) and ultrafine particle lifetimes are ~days, the enhancements observed on ATom in the NH in the fall, winter, and spring cannot
have been from ASM outflow. ATom stratospheric observations did not see evidence of ASM-sources particles (Murphy et al., 2020). Therefore, the ASM is not the cause of the observed NH LMS NPF.

No evidence of larger particles from volcanoes was observed in the NH LMS on ATom, except for in the SH in August (from the 2015 Calbuco eruption) (Murphy et al., 2020). The lifetime of ultrafine particles is shorter than that of larger particles in
the LMS (Supplemental Materials S2), therefore there is no reason to suspect small particles directly emitted from volcanoes would be present in the NH LMS during ATom observations. Volcanoes are, however, another potential source of direct injection of $SO_2$ into the LMS.

Smaller volcanic eruptions are capable of injecting $SO_2$ into the stratosphere. Eruptions contributing substantially to $SO_2$
concentrations and insubstantially to larger particles that would have been detected as volcanic would occur between a few weeks and a couple of months from our measurements for each of the four deployments. Using data from the Multi-Satellite Volcanic Sulfur Dioxide Database Long-Term L4 Global (Carn, 2019), we examine eruptions within 2 months leading up to each ATom deployment with peak plume heights above 7 km (Fig 12, Table S1). Figure 12 shows the latitude and altitude of the recorded plume heights of these eruptions. While the vertical structure of volcanic emission plumes is complex, and such
complexity is not accurately captured by the database used, or in general by satellite retrievals (Carn, 2019), we note that the only eruption to occur between the tropopause height (taken from the 2014 MERRA2 data) and 13 km (the upper limit of ATom observations), or within a few km of these limits, is two days of the December 2016 Bogoslof eruption, totalling 0.0058 Tg $SO_2$ in the NH LMS (Table S1). Total volcanic $SO_2$ emissions above 7 km altitude within two months of observations, for ATom 1-4 respectively were 0.057, 0.0235, 0 and 0.048 Tg in the NH, and 0, 0.002, 0.08 and 0.153 Tg in the SH. The Aoba
eruption in April 2018 was the largest eruption within this time frame, contributing 0.15 Tg to the SH, but the peak plume altitude was 17 km, one of the highest recorded, which may reduce its influence on observations in the LMS which peaked at





around 12 km in the SH in this season. Given the uncertainties in plume structure, it is not possible to estimate the effect these volcanic $SO_2$ emissions had on the observed hemispheric differences in LMS $SO_2$ and small particle concentrations. While it is true that ATom 3 saw the lowest NH volcanic $SO_2$ emissions and the lowest NH small particle number concentrations, in

general the variation between seasons in the NH, and the fact the SH emissions were larger than NH emission in ATom 3 and 4, suggest that this is unlikely to be the cause of seasonally persistent elevated small particle concentrations in the NH LMS. This is supported by the lack of observations of volcanic aerosol of larger sizes in the NH LMS (Murphy et al., 2020), which places stringent limits on timing on volcanic eruptions that can influence $SO_2$ but not larger particles.

The uncertainties in volcano emissions detailed above make any direct comparison of aircraft and volcanic $SO_2$ sources in the LMS impossible, though it seems unlikely that the highly temporally variable $SO_2$ recorded near the LMS from volcanoes could dominate over this strong, seasonally persistent aviation source over four seasons.

**7. Conclusions**

Within the scope of this study, the most likely cause of elevated numbers of ultrafine aerosol in the NH LMS at mid and high

latitudes is aviation, through a combination of direction emission and nucleation in the exhaust plume, and NPF caused by elevated $SO_2$ from aircraft in the background NH LMS. Known uncertainties in volcanic $SO_2$ emissions, the effect on NMVOCs from aircraft and sources of unusual NPF precursors such as halogens and mercury in the NH LMS remain potential causes of the observed hemispheric differences in ultrafine aerosol concentrations that could not be fully ruled-out. A refined bottom-up estimate of aircraft emissions, and further in-situ studies of both aircraft and volcanic emissions are essential to

properly assess their contribution to the observed substantial nucleation mode aerosol concentrations and $SO_2$ mixing ratios in the LMS of the NH.

Ultrafine aerosols co-determine the size distribution particles in the LMS. Because of stratification of air density, most of the mass of the stratospheric aerosol, and hence its radiative effects, is in the LMS, although the mass mixing ratio of particles

maximizes at altitudes > 20 km (Yu et al., 2016). If ultrafine aerosols serve as sites for further condensational growth, they may shift the particle size distribution to smaller sizes in the LMS of the NH. In the LMS light scattering is driven by larger particles, and infrared heating is almost independent of particle size (Murphy et al., 2020). Therefore, a shift of the size distribution to smaller sizes may lead to more warming of the LMS compared to light scattering.

$SO_2$ emissions from aircraft are increasing with time (Lee et al., 2020), and the expectation is for this trend to continue. Furthermore, alternative aircraft fuels are under consideration. Our results show that current levels of aviation have created substantially different conditions in the LMS in the NH compared with the less-anthropogenically-influenced SH, and that the stratosphere system is sensitive to relatively small perturbations. This motivates further study of trace emissions from



alternative aviation fuels, as well as targeted studies to reduce uncertainties on the stratospheric impacts of aviation
currently. We hope these observations act as an early warning to fully understand the effect of aircraft emissions in the
stratosphere before their magnitude increases further.

Climate intervention via injection of $SO_2$ into the stratosphere is being discussed as a potential strategy to temporarily reduce
the effects of anthropogenic greenhouse gases while their emissions are brought under control (Shepherd, 2012;Council,
2015;Keith et al., 2014;MacMartin and Kravitz, 2019). How this would be achieved, and the potential consequences, both
intended effects and side-effects are highly uncertain. A more complete knowledge of the background state of the stratosphere,
and current anthropogenic influence in this region is needed before we can predict the effect of intentional modification on
radiative balance, heterogenous chemistry and circulation. The hemispheric difference in ultrafine aerosol concentrations we
have shown here is an example of how anthropogenic emissions are already modifying the stratosphere. This can be both
studied further to help understand the consequences of any intentional modification of stratospheric composition, and must be
considered in models used to design climate intervention strategies or assess their potential consequences. We must also be
aware that any intentional stratospheric modification will be applied to two very different hemispheres: a largely pristine
southern hemisphere; and an already anthropogenically modified northern hemisphere.

**Data availability:** The ATom dataset is published as Wofsy et al., (2018, https://doi.org/10.3334/ORNLDAAC/1581), and
also available at https://espoarchive.nasa.gov/archive/browse/atom (last access: December 2020). POSIDON data are
available at https://espoarchive.nasa.gov/archive/browse/posidon (last access: January 2021). Specific data and model
outputs presented in this analysis are available at the Oak Ridge National Laboratory (ORNL) Distributed Active Archive
Center (DAAC). Volcano data are from the Multi-Satellite Volcanic Sulfur Dioxide (SO2) Database Long-Term L4 Global,
described in Carn (2019) are available at https://disc.gsfc.nasa.gov/datasets/MSVOLSO2L4_3/summary (last access:
November 2020). The CEDS emissions database is documented in Hoesly et al., ( 2017, https://doi.org/10.5194/gmd-11-
369-2018), with details available at http://www.geosci-model-dev-discuss.net/gmd-2017-43/ (last access: November 2020).
MERRA2 data products can be obtained from https://disc.sci.gsfc.nasa.gov and the 2014 zonally averaged tropopause
heights and potential temperatures are included in the ORNL DAAC dataset related to this analysis.

**Author contributions:** Data collection and analysis on ATom was carried out by CJW, AK and CAB (aerosol size
distributions), AR ($SO_2$), KDF, GPS and DMM (single particle composition), JP, CT, IB, and TR (ozone), GSD and JPD
(water vapour), TPB (meteorological and global positioning), DRB (trace gases), and MD and BW (cloud properties).
POSIDON $SO_2$ data were collected by AR. EAR ran the ATom back-trajectories. JK ran the MAIA box model and KDF made
the thermodynamic calculations. CJW wrote the manuscript, with input from all co-authors.





**Competing interests:** The authors declare that they have no conflict of interest.

**Disclaimer:** The contents do not necessarily represent the official views of the University of Colorado, NOAA, the University of Vienna, or of the respective granting agencies. The use or mention of commercial products or services does not represent an endorsement by the authors or by any agency.

**Acknowledgements:** We thank K. Aikin, M. Richardson, H. Bian, J. Wilson and D. Axisa for contributions to this analysis,
and the ATom leadership team, science team and crew for contributions to the ATom measurements.

**Financial support:** This work was supported by the National Aeronautics and Space Administration's Earth Venture program through awards NNX15AJ23G and NNH15AB12I and by NOAA's Health of the Atmosphere and Atmospheric Chemistry, Carbon Cycle, and Climate programs. Agnieszka Kupc was supported by the Austrian Science Fund FWF's Erwin Schrodinger Fellowship J-3613. Bernadett Weinzierl and Maximilian Dollner were supported by European Research Council
(ERC) under the European Union's Horizon 2020 research and innovation framework program under grant 640458 (A-LIFE) and by the University of Vienna. We thank the NASA Upper Atmosphere Research Program for providing funding to allow for the POSIDON deployment to Guam.

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

**Figures**





**Figure 1: ATom flight locations.** Flight tracks from ATom in all four seasons as a function of latitude and longitude (a), and altitude and latitude (b-e). Sampling in the LMS (O3 > 250 ppb, relative humidity (RH$_w$) <10 %, are highlighted in light-coloured symbols. World map was made with Natural Earth Free Vector and Raster Map Data. http://www.naturalearthdata.com (accessed 10 December 2015).



**Figure 2: Hemispheric Differences in Particle Number and Concentrations.** Histograms of the total number of aerosol between 3 and 4500 nm in the LMS (ozone 250-400 ppbv, altitude > 8 km) for the SH and NH) for all for ATom deployments (a-d), by season. Fractional excess of mean particle number in the NH compared to the SH (($N_{NH}$-$N_{SH}$)/$N_{NH}$) in each season as a function of particle diameter (e).




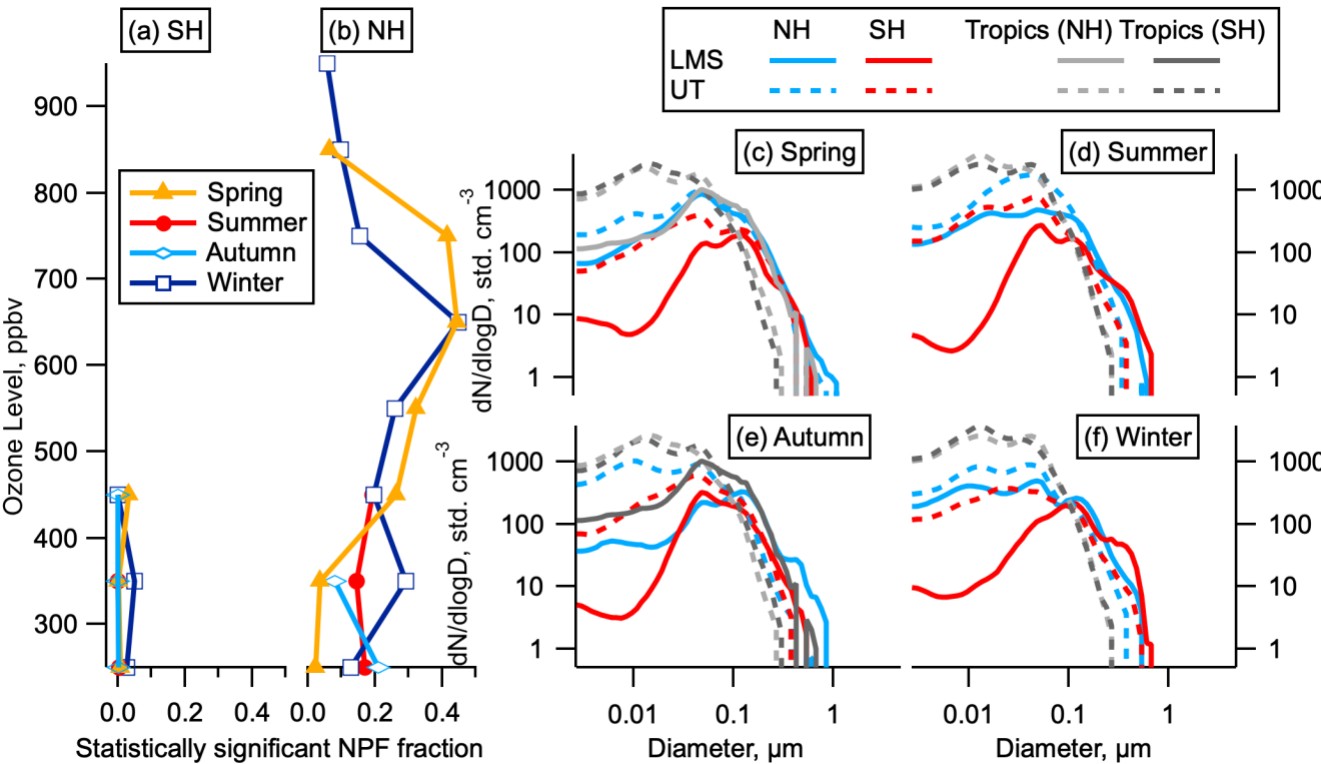

**Figure 3: Statistically significant NPF and Size Distributions by hemisphere.** (a) and (b) The fraction of data indicating statistically significant NPF as a function of ozone for each hemisphere and season. Data is limited to altitudes above 8 km. (c-f) Average size distributions in the LMS and UT in the NH (latitudes above 30°), SH (latitudes below -30°), and tropics (-30° to 30°) by season. The LMS was not sampled in the tropics except for in May. Tropics are shown both for the month corresponding to the NH season (dark grey) and the SH season (light grey) e.g. in Spring the tropical May observations are shown in light grew and the tropical October observations are shown in dark grey.



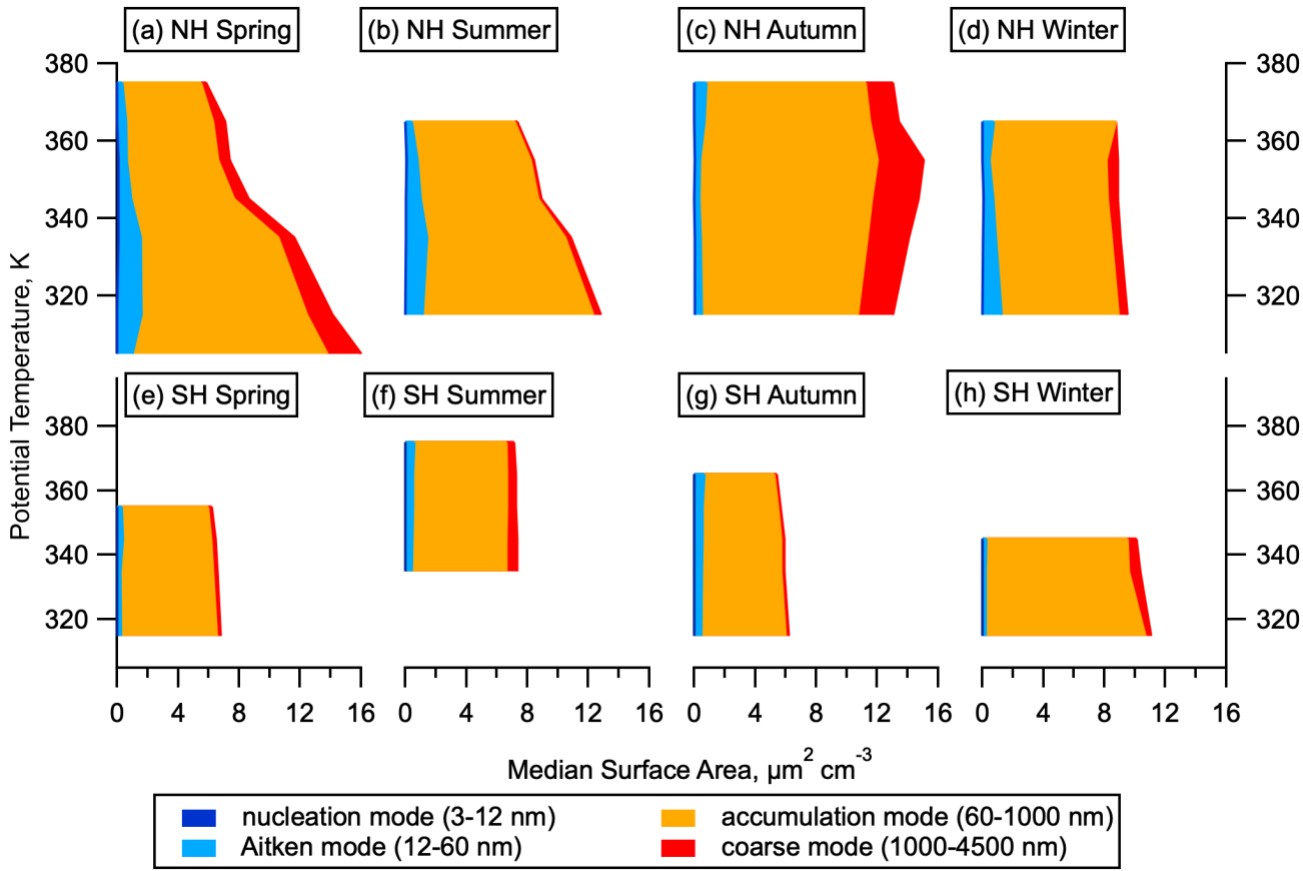

**Figure 4: Aerosol surface area in the LMS.** Aerosol surface area in the LMS as a function of potential temperature, separated by mode
(colours) and season for the NH (a-d and SH (e-h)). Mode size ranges are 3-12 nm (nucleation), 12-60 nm (Aitken), 60-1000 nm
(accumulation), and (1000-4500 nm) coarse.





**Figure 5: SO₂ in the LMS.** (a) Mixing ratios of gas phase SO₂ in the LMS and LMS (450>O3>200 in the SH (open blue) and NH (filled red).) (b) Median and interquartile range of SO₂ by latitude in May for the LMS (red) and UT (Blue) in May. UT is defined as altitudes over 8 km and O₃ < 250 ppbv or RHw > 10%. SO₂ observations at mixing ratios below 100 pptv are only available for the May 2018 deployment. (c) Tropical UT and LMS SO₂ mixing ratios as a function of potential temperature from POSIDON, Oct 2016 and from ATom, May 2018.





**Figure 6: Aerosol correlations with SO₂.** Number concentration dependence on SO₂. Number concentration as a function of SO₂ mixing ratios in May (high resolution SO₂ data was not available in other seasons) for the Northern and Southern Hemispheres (red cross and blue circles respectively). Data are divided into four modes by size: (a) nucleation (3-12 nm); (b) Aitken (12-60 nm); (c). accumulation (60-1000 nm); and (d) coarse (1000-4500). Observations where SO₂ is below the limit of detection (1 pptv) are not shown.



**Figure 7: Box modelling and back trajectories**. 48 hour back trajectories from selected observations in the NH LMS in May. Geographical location of back trajectories (lines) and observations (circles, coloured by ozone level) are shown in a. Black lines are trajectories where the relative humidity (RH$_w$) was mostly below 10 %, indicating stratospheric conditions. Light blue lines are trajectories where the RH$_w$ was often above 10 %, indicating tropospheric conditions. RH$_w$, pressure, sulphuric acid, and SO$_2$ (from initial 20 pptv SO$_2$) are shown in b, c, d and e respectively. Modelled number concentration of 2.7 nm particles on the high RH$_w$ (blue) and low RH$_w$





(black) trajectories are shown for initial 20 pptv SO$_2$ and starting condensation sink (CS) of 3 x 10$^{-4}$ s$^{-1}$ (f), initial 40 pptv SO$_2$ and starting condensation sink of 3 x 10$^{-4}$ s$^{-1}$ (g), and initial 20 pptv SO2 and starting condensation sink of 1 x 10$^{-4}$ s$^{-1}$ (h). World map was made with Natural Earth Free Vector and Raster Map Data. http://www.naturalearthdata.com (accessed 10 December 2015).

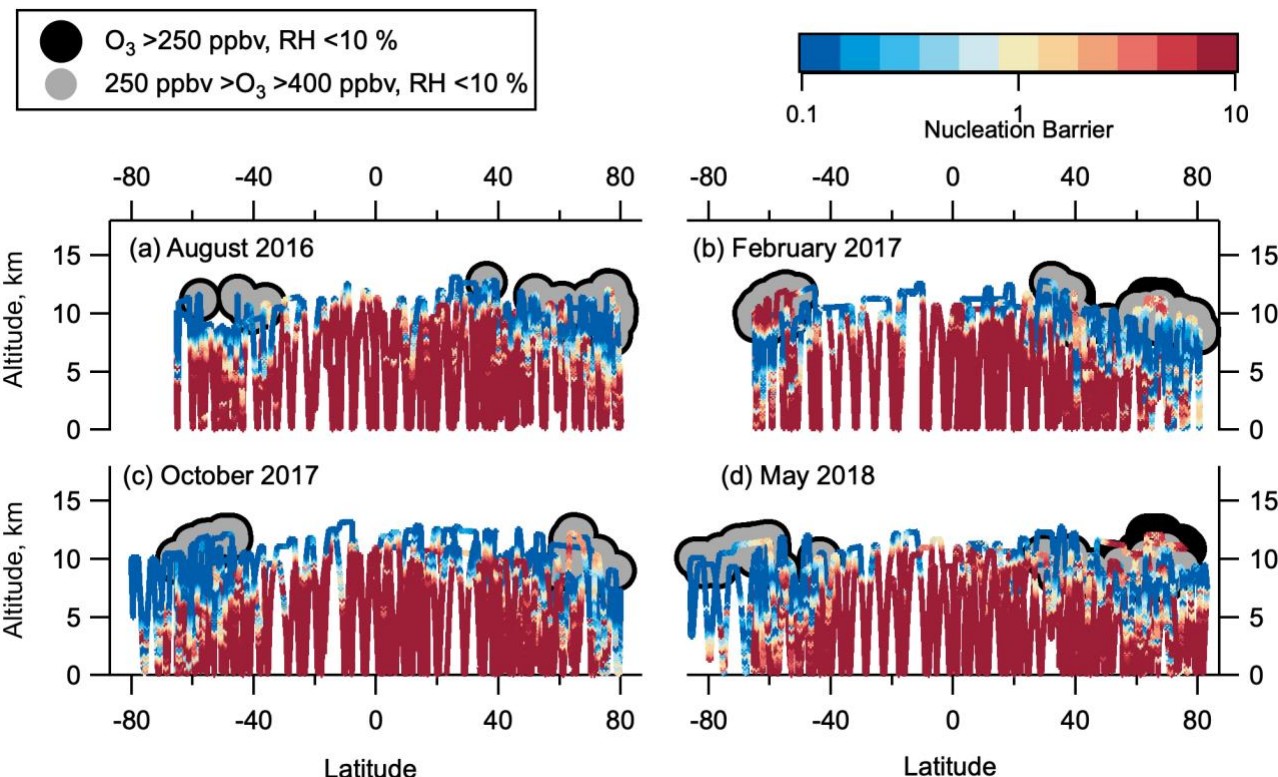

995

**Figure 8: Thermodynamic barriers to nucleation.** Thermodynamic nucleation barrier calculated of negative ion H2SO4-H2O clusters with p(H$_2$SO$_4$) = 0.1 pptv for each season of ATom observation. Values >= 10 (dark red) are insurmountable and negative ion H2SO4-H2O NPF cannot occur. Barrier values <=1 (yellow-blue) means that nucleation occurs unimpeded by a thermodynamic barrier if sufficient precursor vapor is present. Barrier between 1 and 10 (yellow-red) means nucleation can occur, but there is some thermodynamic barrier that needs to be overcome. Regions highlighted in grey have been identified as LMS (O3 > 250 ppbv, RHw < 10%).





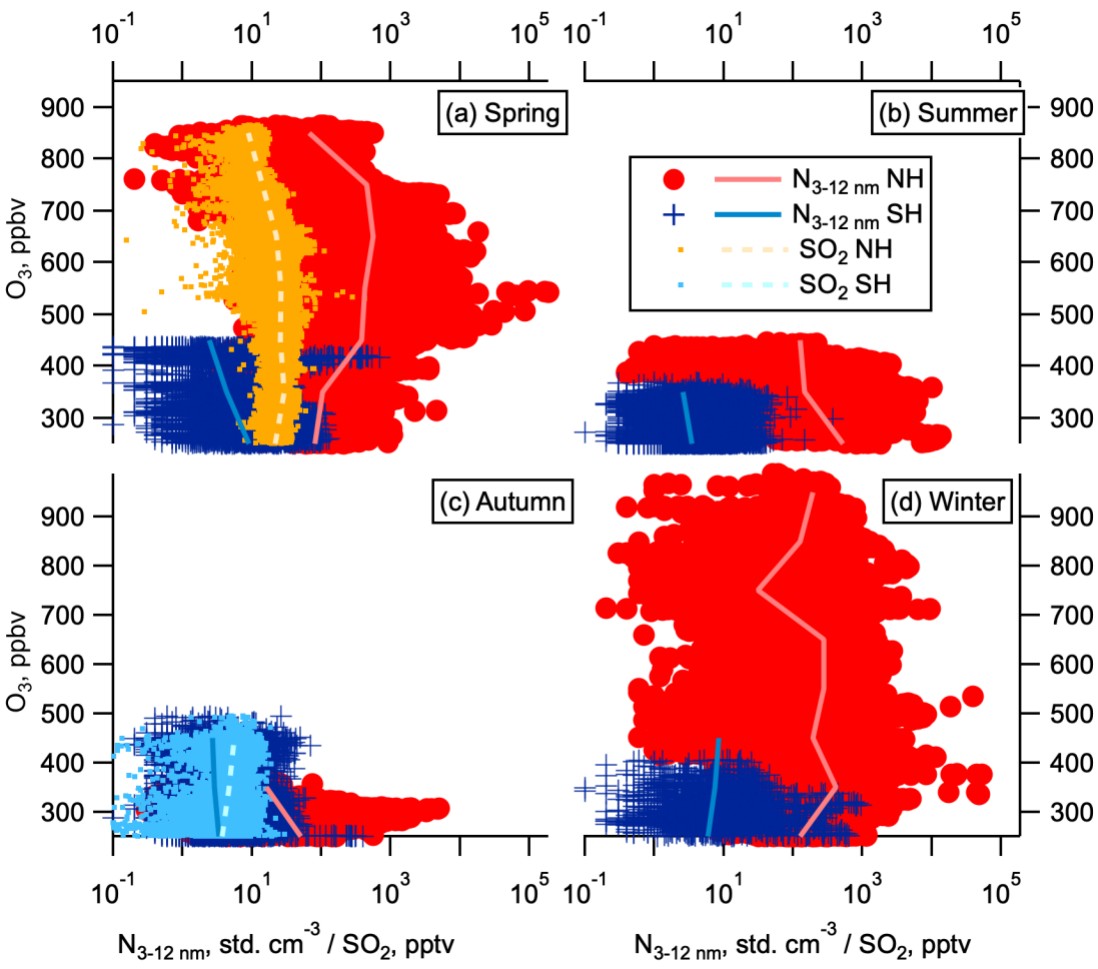

**Figure 9: LMS aerosol and SO₂ variation with ozone.** Nucleation mode aerosol number concentration and SO₂ concentration (shown only in May as this was not measured for other seasons) as a function of ozone in the LMS in each hemisphere for each season (a-d).





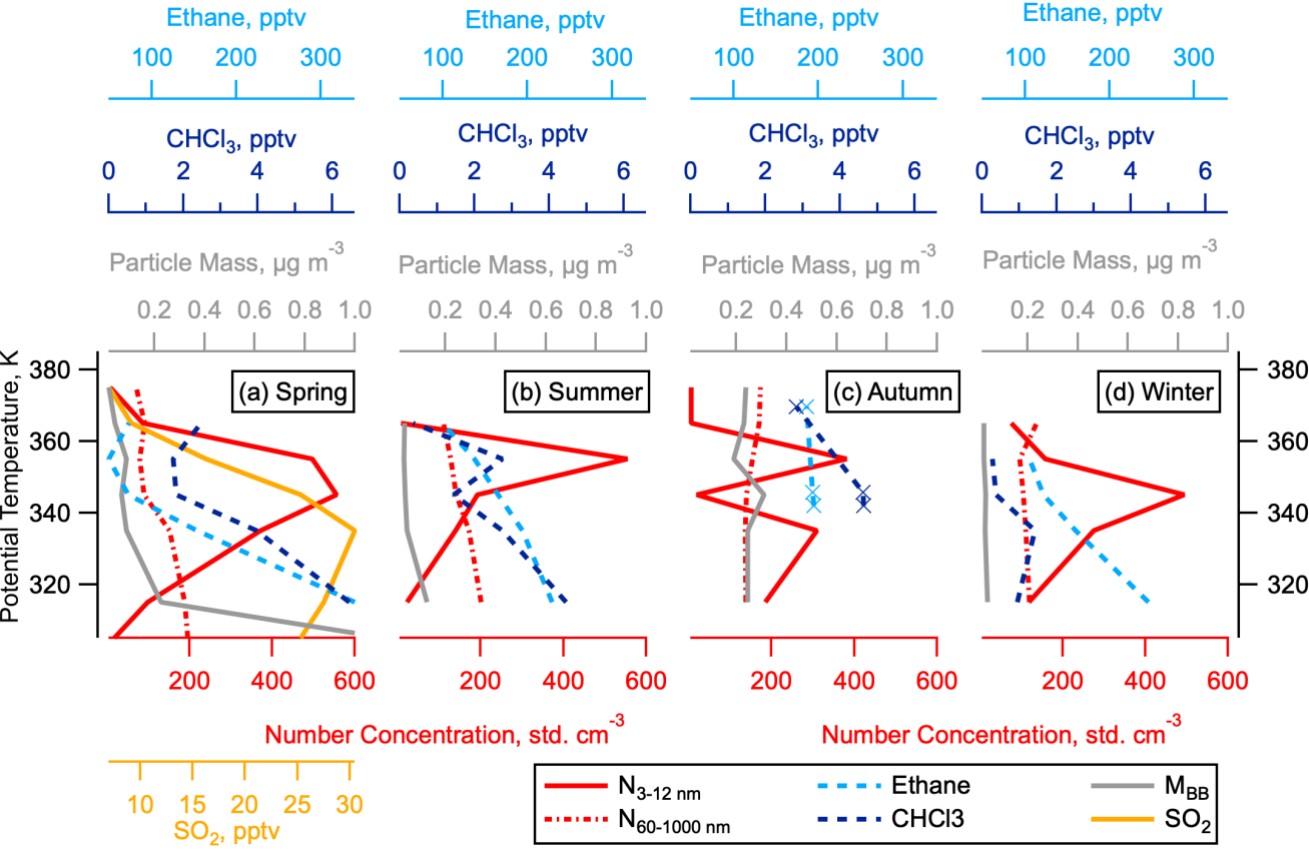

1005 **Figure 10: Vertical Profiles of small particles and tropospheric tracers in the NH LMS.** 50th percentiles of number concentrations of nucleation mode (3-12 nm) and accumulation mode (60-1000 nm) particles, mass concentration of particles from biomass burning, concentrations of ethane and Chloroform (CHCl3), as a function of potential temperature in the NH LMS. The 50th percentile of the measured SO2 concentration is given in spring only, as these measurements were not taken in other seasons.





**Figure 11: CEDS 2014 aircraft emissions SO₂ flux and calculated LMS concentrations.** (a) SO₂ flux from aircraft from CEDS emissions database for May 2014, between 11.6 and 12.2 km altitude on a log-colour scale as a function of latitude and longitude, with ATom flight



tracks from all seasons overlaid with LMS nucleation mode number concentrations measurements coloured on a log-scale for comparison. (b) SO$_2$ flux (in kg m$^{-2}$ s$^{-1}$) from aircraft from CEDS emissions database zonally average for the whole year of 2014 as a function of altitude and latitude with the tropopause height from MERRA2 2014 in red. (c) Calculated, zonally averaged average 2014 SO$_2$ concentrations plotted on a linear colour scale for altitudes >8.5 km from these aircraft emissions, assuming a 1-month lifetime of SO$_2$ in the LMS, as a function of latitude and altitude with the tropopause height from MERRA2 2014 in red. (d) Median SO$_2$ calculated as for panel (c) but scaled by a factor 1.23 to take into account increases in air-traffic between 2014 and 2018(Lee et al., 2020). MERRA2 data are used to convert CEDS emissions onto a potential temperature scale, and average profiles above 20° latitude are compared with the median ATom LMS SO$_2$ observations in each hemisphere from May 2018. 25-27[th] percentiles are given in shaded areas of the CEDS emissions, and error bars for ATom observations. World map was made with Natural Earth Free Vector and Raster Map Data. http://www.naturalearthdata.com (accessed 10 December 2015).

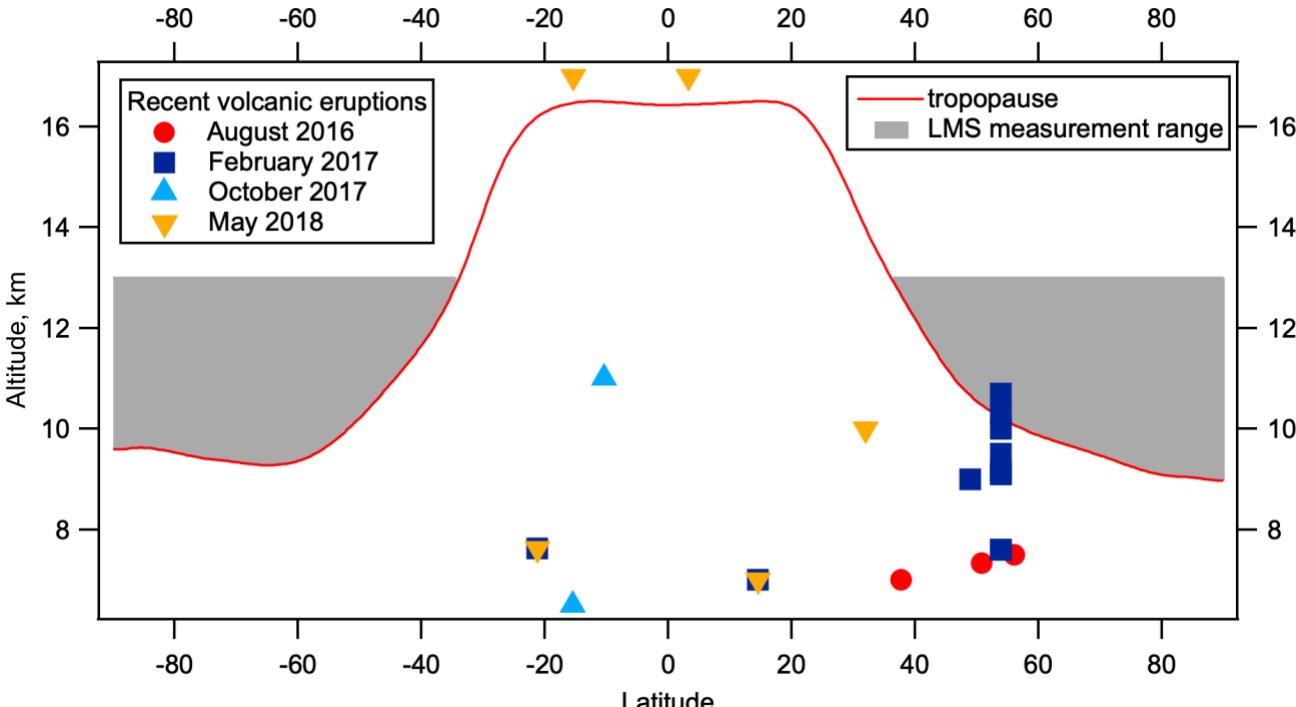

**Figure 12: Volcanic eruptions within two months of ATom observations near the LMS.** Volcanic eruptions plotted by latitude and plume altitude for 2 months prior to each ATom deployment where the plume altitude is above 7 km. Eruptions are shown with colors/symbols corresponding to the relevant ATom deployment. Tropopause height as a function of latitude from 2014 MERRA2 data is shown in red, with the area between the tropopause and 13 km highlighted in grey. Details of these eruptions can be found in Table S1. Volcano data are from the Multi-Satellite Volcanic Sulfur Dioxide (SO2) Database Long-Term L4 Global.