# Peer review of "Large hemispheric difference in ultrafine aerosol concentrations in the lowermost stratosphere at mid and high latitudes"

_Atmospheric Chemistry and Physics, 2021_

## Author Response (AR1)

**Responses to Referees and Editor on**

**"Large hemispheric difference in nucleation mode aerosol concentrations in the lowermost stratosphere at mid and high latitudes"**

**Christina J Williamson et al.**
**2021-03-31**

**Responses to Referee #1**

We thank the referee for their insightful comments on the manuscript, which have improved the quality and clarity of the science presented. We have addressed the comments as detailed here below.

*The size range 3-12 nm is called sometimes ultrafine particles, sometimes nucleation mode particles in the paper. I would strongly recommend avoiding the term ultrafine particles in this context, as the vast majority of air pollution scientists use this term for the whole <100 nm particle population. The authors have also applied the size ranges 3-12 nm and 12-60 nm for the nucleation and Aitken mode, respectively. In most studies conducted in the lower troposphere, the border between the nucleation and Aitken mode has been assumed to be somewhere in the range*
*20-30 nm, while the Aitken mode has been assumed to extend up to 90-100 nm. The lower size ranges for these modes applied here are acceptable because the whole particle population seems to be shifted to smaller sizes, possibly due to lower concentration levels of aerosol precursors, compared with the lower troposphere. However, due to the somewhat unusual definitions of nucleation and Aitken mode size ranges, I suggest that the author add a couple of lines into the text to explain why they used such definitions for the nucleation and Aitken modes.*

The referee is correct that our referral to 3-12 nm particles at "ultrafine" is out of step with the community at large, and revise the manuscript to adopt their recommendation of sticking with the defined term "nucleation mode". These changes are made in the following lines:

1 – title, "ultrafine aerosol concentrations" to "nucleation mode aerosol concentrations"
– "mode of ultrafine aerosol" to "mode of aerosol smaller than 12 nm"
– "ultrafine particles" to "nucleation mode particles" (both times)
90, 92, 95, 102 – "ultrafine" to "nucleation mode"
– "nucleation and/or growth of ultrafine aerosol" to "nucleation and/or growth of aerosol"
276 - "ultrafine" to "nucleation mode"
– "ultrafine particles lifetimes" to "nucleation mode particle lifetimes"
– "lifetime of ultrafine particles" to "lifetime of nucleation mode particles"
434, 439, 469, 473, 478, 480, 499 - "ultrafine" to "nucleation mode"

Regarding the boundary between Aitken and Nucleation mode being somewhat low compared to the lower troposphere standard, we agree with the referee that indeed the lower concentration of precursors may well be a factor here. We also use this size range because the ATom dataset reports modes with these size limits, a conscious decision to allow direct comparison with observations by Anthony Clarke et al. of new particle formation and CCN in the remote marine
atmosphere (Clarke et al., 1999;Clarke et al., 2013;Clarke and Kapustin, 2002;Clarke et al., 1998). This size range has subsequently been used in a number of publications discussion ATom observations (Brock et al., 2019;Kupc, 2020;Williamson et al., 2019). We agree that it is worth adding further explanation of this in the manuscript and have done so at line 152 with the following statement:

"In this analysis we divide the aerosol size distribution into four modes: nucleation mode from 3-12 nm, Aitken mode from 12-60 nm, accumulation mode from 60-1000 nm, and coarse mode above 1000 nm. We note that the 12 nm cut-off between nucleation and Aitken modes is defined at a smaller diameter than is often used for aerosol studies in the lower troposphere, and make this choice to allow for more direct comparison with previous datasets over the remote Pacific and Atlantic oceans (Clarke and Kapustin, 2010;Clarke et al., 1999;Clarke et al., 2013;Clarke and Kapustin, 2002;Clarke et al., 1998), and for consistency with the ATom data archive (Wofsy et al., 2018) and other published works analyzing ATom size distributions (Brock et al., 2019;Kupc et al., 2020;Williamson et al., 2019)."

*Somewhat related to the previous comment, it is not a good practice to talk about small particles (e.g. title of section 3) or larger particles (lines 438 and 481) without specifying what is exact meant by small or larger here.*
        We agree that our use of unquantified "small" and "large" descriptors of particles was poor practice and have amended this throughout the manuscript to be more precise. Changes made, and the corresponding line numbers are as follows:

57, 74, 174, 183, 208, 221, 235, 262, 265, 290, 294, 314, 335, 426, 427, 440, 458, 459, 461, 1005 – "small" to "nucleation mode"
        195, 197, 199, 338 – "small" to "nucleation and Aitken mode"
        356 – "small particle emissions" to "particle emissions"
        47 – "larger particles" to "particles without details of the nucleation mode"

196 – "larger" to "accumulation and coarse mode"
        251 – "larger particles" to "particles with diameters between 350 and 600 nm"
        312 – "larger particles of tropospheric origin" to "accumulation mode particles of tropospheric origin"
        438, 439, 445,462, 463 – "larger" to "accumulation mode"

        *In several places of the text, the authors talk about correlation and even its character (significant, slight). Technically, correlation is an exact statistical quantity, which should be used just based on visual the appearance on how two variables seem to be connected with each other. I recommend using some other term than correlation in the text or, alternatively, to calculate the actual correlation coefficient and its level of significance.*

With the discussion of correlation as it relates to nucleation mode number concentrations and SO2 concentrations, the referee is correct. We have therefore modified Fig. 6 to show log-log fits between number concentrations and $SO_2$ with correlation evaluated using r-squared values. Please note, in order to remove the effect of instrumental noise on correlations, we have analyzed the $SO_2$ and nucleation mode data to 0.1 Hz (the original fig. 6 showed 1 Hz data) following the analysis of noise in Williamson et al. (2019).

        The new fig. 6 and caption are as follows

[Figure]

Figure 1: Aerosol correlations with $SO_2$. Number concentration dependence on $SO_2$ at 0.1Hz time resolution to reduce the effect of atmospheric and instrumental noise following Williamson et al. (2019). Number concentration as a function of $SO_2$ mixing ratios in May (high resolution $SO_2$ data was not available in other seasons) for the Northern and Southern Hemispheres (red circles and blue dots respectively). Data are divided into four modes by size: (a) nucleation (3-12 nm); (b) Aitken (12-60 nm); (c). accumulation (60-1000 nm); and (d) coarse (1000-4500). Log-log fits are given by the sold lines. Panel (e) shows $R^2$ values for these fits. Observations where $SO_2$ is below the limit of detection (1 pptv) are not shown.

Line 220 has been changed to read

"Number concentrations of nucleation and Aitken mode particles show some correlation with $SO_2$, whereas number concentrations of accumulation and coarse mode particles show little to no correlation with $SO_2$ (Fig. 6)"

Regarding correlation with O3, we have changed how we describe the SO2 behaviour at line 322 to the following:

"We do not observe $SO_2$ increasing with $O_3$ in the NH LMS (Fig. 9a), and therefore conclude that the NH excess $SO_2$ is not from OCS oxidation. $SO_2$ does increase with $O_3$ in the SH LMS (Fig. 9c), suggesting OCS oxidation may be contributing to these lower, background levels of $SO_2$."
We also note that we had mistakenly referred to Fig. 7 here, and have therefore corrected it to Fig. 9.

We also noticed some repetition of text between line 319 and 333. We have corrected this by deleted the repeated text from line 325-329.

We have corrected the reference to no correlation between nucleation mode number concentrations and ozone from lines 330-332 to the following:
"NPF can take place in descending air in the polar vortex in winter, and quasi-horizontal transport is a potential pathway for these particles to then enter the LMS. However, similar to $SO_2$, we do not observe nucleation mode particle concentrations increasing with O3 is (Fig. 9), suggesting they do not originate deeper in the stratosphere."

Where we previously talked about lack off correlation, and slight anti-correlation between nucleation mode and biomass burning particles at line 427, we have corrected this to read: "Nucleation mode and biomass burning particles in the NH LMS did not follow the same trends with potential temperature (Fig. 10), which suggests that pyro-convection is not the major source of the nucleation mode particles we observed. In October, there is some indication in the vertical structure of nucleation mode number concentrations being suppressed at higher biomass burning particle mass concentrations, and enhanced at lower biomass burning particle mass concentration."

*Line 361: The lifetime of nucleation mode particles depends both on the mean size of these particles and on the properties of the pre-existing particle population at sizes larger than the nucleation mode. Considering that both of these quantities are probably quite variable in the LMS, and especially quite different between NH and SH (as discussed in the supplementary material and illustrated in Figure S3), the authors should better justify the use of a single lifetime of 2 days for nucleation mode particles in their calculations.*

We are grateful to the referee for drawing attention to this problem. On further examination, we concluded that calculating an overall lifetime of nucleation mode aerosol in the SH LMS, taking into account the orders of magnitude difference expected between plume (days) and non-plume (weeks-months) lifetimes, is not possible. For illustrative purposes we decided that potential concentrations calculated using a 2-day lifetime, and thus representing a lower-limit, were useful, and rewrote the paragraph beginning at line 360 as follows:

"While particles lifetimes in non-plume LMS conditions are expected to be longer in the SH than the NH, nucleation mode particle lifetimes within aircraft plumes have been shown to be around

2 days (Schroder et al., 2000). The plume and non-plume lifetimes in the NH LMS are similar enough to justify the use of a 2-day lifetime in this region (Supplemental Materials S2). In the SH LMS, there is a large difference between in-plume and non-plume lifetimes (from days to months). We use a 2-day lifetime here in the SH for illustrative purposes, but note that this is an over-estimate, and thus the resulting concentration will represent an underestimate. Calculating the volume of the region defined above as the LMS to be 3.12 x $10^{17}$ m$^3$ and 3.65 x $10^{17}$ m$^3$ for the NH and SH respectively, we get ambient concentrations of nucleation particle emitted by aircraft in the LMS of 110 cm$^{-3}$ in the NH and a lower limit of 0.26 cm$^{-3}$ in the SH. Taking an average temperature and pressure of 220 K and 300 hPa for the LMS regions, and converting to standard temperature and pressure (STP), this gives maximum concentrations of 300 std. cm$^{-3}$ for the NH and a lower limit of 0.71 std. cm$^{-3}$ for the SH. Minimum aircraft emissions of nucleation mode aerosol reported by Brock et al. (2000) are a factor of 10 lower than the maximum we have used here, leading to concentrations of 30 and 0.07 std. cm$^{-3}$ for the NH and SH respectively."

*I am basically fine with the way this paper discusses and speculates about the particle origin in the study regions, including the mechanism of new particle formation. The only thing that could be improved the analysis of probability of (ion-induced) water-sulfuric acid nucleation is evaluated here based on the model framework developed by Kazil, Lovejoy and co-workers 10-20 years ago. Since then, detailed laboratory data on the same nucleation mechanisms has been obtained e.g. in CLOUD experiments, and these data have also been included in nucleation parameterizations. I am not saying that the authors should redo their calculations, but they could shortly discuss whether their conclusions are also consistent with this most up-to-date information on atmospheric water-sulfuric acid nucleation at cold temperatures.*

While the CERN CLOUD collaboration has, to the best of our knowledge, not published thermochemical data (enthalpy and entropy of formation) of neutral and charged $H_2SO_4/H_2O$ clusters from their chamber experiments, which would allow a direct comparison with the experimental thermochemical data used in MAIA (Curtius et al., 2001;Lovejoy and Curtius, 2001;Froyd and Lovejoy, 2003;Hanson and Lovejoy, 2006), two indirect comparisons exist in the literature. We have added the following passage to the description of MAIA at line 173:

Aerosol nucleation rates calculated  from the experimental thermochemical data of neutral and charged $H_2SO_4/H_2O$ cluster formation that are used in MAIA (Kazil and Lovejoy, 2007) compare well with neutral and charged $H_2SO_4/H_2O$ nucleation rates measured in the European Organization for Nuclear Research (CERN) Cosmics Leaving Outdoor Droplets (CLOUD) chamber (Kirkby et al., 2011). Global model simulations, either using a parameterization of neutral and charged $H_2SO_4/H_2O$ nucleation based on the CERN CLOUD chamber measurements, or nucleation rates calculated from the experimental thermochemical data used in MAIA (Kazil et al., 2010) show a good agreement in the global mean profile of total (> 3 nm) aerosol concentration (Määttänen et al., 2018).

*Line 83: The authors could add the study by Sipila et al. (2016, Nature 537, p. 532), because it is the very first study in which iodine compound have been measured in molecular clusters associated with new particle formation in a coastal atmosphere.*

We thank the referee for this suggestion, and have added the reference accordingly.

*Lines 193-194: Stating that SO2 concentration measurements are sensitive to <100 pptv does not really tell anything useful to the reader. What is the actual detection limit of the instrument under the operating conditions of this study, and reliable are the SO2 of just a few pptv reported in many of the figures?*

The referee makes a good point. The reference to concentrations <100 pptv was made to differentiate between measurements of SO2 made on ATom by the laser induced fluorescence (LIF) instrument, which have a detection limit of 1pptv, and are thus highly relevant to this discussion of NPF in very remote regions, and another SO2 measurement made on ATom, with detection limits > 100 pptv, which are more pertinent to analyses of the more polluted regions. This was poorly explained, and we have made the following changes to the text:

Line 115 in the method sections has been changed from

"SO$_2$ observations sensitive at <100 parts per trillion by volume (pptv; nmol mol$_{-1}$) were made on the fourth set of flights (May 2018) using laser-induced fluorescence techniques (Rollins et al., 2017)."

to

"SO$_2$ observations with a detection limit of 1 parts per trillion by volume (pptv; nmol mol$^{-1}$) were made on the fourth set of flights (May 2018) using laser-induced fluorescence techniques (Rollins et al., 2017)"

Lines 193-194 have been changed from

"Measurements of SO2 mixing ratios with sensitivity to concentrations < 100 pptv were made on the fourth ATom deployment in May 2018."

to

"Measurements of SO$_2$ mixing ratios with pptv sensitivity were made on the fourth ATom deployment in May 2018."

*Line 243: It seems that something is missing from this text: ...it is conceivable the low...The same text is repeated on lines 319-323 and lines 325-329, except that they refer to a different figure.*

Thank you for spotting this! We have corrected it as noted above in the discussion of removing references to "correlation" where supporting evidence was lacking.

**Unsolicited corrections**

We would like to alert the referee to some unsolicited corrections we would like to make to the manuscript that came up in the course of addressing referee comments. These are detailed here below. We also would like to alert the referee to proposed corrections in our response to referee #2.

Extraneous comma removed line 57

For clearer reading, Line 62 change from
"Ammonia and amines have been shown to contribute to NPF"
to
"Ammonia and amines have been shown to be involved in NPF"

Line 82: "lowermost stratosphere" changed to "LMS" for consistency.

Line 123 "CH3Cl" corrected to "CH$_3$Cl"

Line 127
"using Fuchs expression for the coagulation rate coefficient(Seinfeld and Pandis, 2006)"
corrected to

"using the Fuchs expression for the coagulation rate coefficient (Seinfeld and Pandis, 2006)"

Line 141 changed from "in' to "by" in "This stratospheric definition is consistent with that used by Murphy et al. (2020)".

We noted a missing parenthesis on line 195 and so added this.

Line 227, alteration made in order to read better from
"20 of the 55 trajectories experience more humid air, indicative of UT or tropopause conditions."
to
"The other 20 trajectories experienced more humid air, indicative of UT or tropopause conditions."

The full description of Asian Summer Monsoon (ASM) has been placed on the first usage at line 253 instead of where it was mistakenly put on line 424.

Line 309 "Because the lifetime of these particles is ~ days" changed to "Because the lifetime of
these particles is on the order of days"

Line 341 "from" added to "through NPF resulting from $SO_2$ oxidation"

Line 402 "higher observed concentrations … are also likely" corrected from "is also likely"

Line 405 "we get a total flux" change to "we determine a total flux"

Line 435 "ASM-sources particles" corrected to "ASM-sourced particles"
Line 465 "volcano emissions" changed to "volcanic emissions"

Line 451 "the only eruption to occur between the tropopause height: corrected to "the only eruption to reach between the tropopause height"

For clarity, line 495 has been rewritten from
"How this would be achieved, and the potential consequences, both intended effects and side-effects are highly uncertain."
to
"How this would be achieved, and the potential consequences of such actions (both the intended effects and any unintended side-effects) are highly uncertain."

We noted in the SM line 44 a forward slash had accidentally been used instead of a period, and have corrected this.

Additional references have been added at line 53 and the order of references changed to
"New particle formation (NPF) has been well documented in a variety of locations in the planetary boundary layer and free troposphere (Clarke et al., 1998;Clarke et al., 2013;Kulmala et al., 2013;Williamson et al., 2019)."

Similarly, at line 284 which now reads

New particles have previously been shown to … form in the tropical UT (Clarke et al., 1998;Clarke and Kapustin, 2002;Clarke et al., 2013;Williamson et al., 2019)

Figure 3: "level" removed from a,b y-axis and corrections made to caption text: "except for" changed to "except", "grew" changed to "grey"

Figure 4: y-axes for NH and SH changed to be the same range to make comparison easier

Figure 8: greater than and less than symbols in legend corrected

Figure S3. We have added a more through explanation of the figure and how this was calculated from the data in the caption, and included an example mass spectrum to illustrate this. The new proposed figure and caption are as follows:

[Figure]

*Figure S1: Aerosol acidity in the LMS. a) An example negative ion spectrum of an acidic sulphate particle. This spectrum is from a 0.39 μm diameter particle in the stratosphere at 12.2 km and 310 ppbv of $O_3$ on 20171009. Laboratory calibrations show that the $H_2SO_4 \bullet HSO_4^-$ peak is very small or non-existent for particles composed of ammonium sulphate and the cluster ion peak increases with acidity until it is a large peak for nearly pure sulfuric acid. b) The bars for the NH (left) and SH (right), separated by season, show the average ratio of the size of the cluster peak at m/z 195 to the sum of the peaks at m/z 195 and 97. The averages are for particles when $O_3$ concentrations were 250 to 350 ppbv in the stratosphere. The averages are also for particles between 0.35 and 0.6 μm diameter because in the stratosphere most particles of that size originated in the stratosphere. Lab calibrations of particles composed of $(NH_4)_{0.25}H_{1.75}SO_4$ had a negative ion ration m/z 195/(97+195) of 0.034, therefore we consider ratios higher than this (more acidic) to contain less than 0.25 mole fraction ammonium. Rather than analyzing possible differences in the acidity with season, here we emphasize that stratospheric particles in all seasons and both hemispheres are highly acidic. This sets limits on the possible concentration of gas phase ammonia. Calculations of uptake from the gas phase show that a continuous 1 pptv of gas phase ammonia could add 0.25 mol fraction ammonium to sulfuric acid particles in less than a week.*

**Responses to Referee #2**

We thank the referee for their thorough and pertinent review. We have addressed the comments as detailed here below and are grateful from the improvement this has made to the manuscript.

*The description of how box modelling is done needs to be detailed better. In its current form it is really difficult to understand how to simulations are done. The reader is pointed to Kupc et al., 2020 for the description of the box model setup. However, in that paper both MAIA and TOMAS models are used, so it would be easier for the reader to understand the modelling part if it was briefly summarized in this paper.*

We have expanded the description of MAIA at line 166 to read as follows:

"To more quantitatively assess the effects of thermodynamics on NPF in the LMS, box modeling is performed using the Model of Aerosols and Ions in the Atmosphere (MAIA), (Lovejoy et al., 2004;Kazil and Lovejoy, 2007;Kazil et al., 2007). MAIA describes the oxidation of $SO_2$ to gaseous $H_2SO_4$, the nucleation of neutral and negative $H_2SO_4$-$H_2O$ clusters, aerosol growth by 340 sulfuric acid condensation/evaporation, and particle coagulation. The production rate of $H_2SO_4$ is calculated assuming that the reaction of $SO_2$+OH is the rate limiting step of the oxidation of $SO_2$ to form $H_2SO_4$ (Lovejoy et al., 1996). Nucleation is described with laboratory thermochemical data for $H_2SO_4$ and $H_2O$ uptake and loss by small neutral and negative clusters (Curtius et al., 2001;Lovejoy and Curtius, 2001;Froyd and Lovejoy, 2003;Hanson and Lovejoy, 2006). The 345 thermochemical data for uptake and loss of $H_2SO_4$ and $H_2O$ by large sulfuric acid aerosol ($\gg$ 5 sulfuric acid molecules) are based on the liquid drop model and $H_2SO_4$ and $H_2O$ vapor pressures over bulk solutions. These were calculated with a computer code (provided by S. L. Clegg, personal communication, 2007) which adopts experimental data from Giauque et al. (1960) and (Clegg et al., 1994). The thermochemical data for intermediate sized particles are a smooth 350 interpolation of the data for small and large aerosol particles. The model uses 20 linear bins in which $H_2SO_4$ content increases by 1 molecule per bin, and 50 geometric bins in which $H_2SO_4$ content increases by a factor of 1.45 per bin, covering a dry (312.15 K, 10% RH) particle diameter range of ~0.5–800 nm.

MAIA operates along trajectories with changing pressure, temperature and relative humidity (Kazil and Lovejoy, 2007) in the temperature range 180-320 K and the relative humidity range 1-101 %, which includes upper troposphere conditions. MAIA parametrizes the OH diurnal cycle as a half-sine centered around noon with a prescribed noon OH concentration, while setting the nighttime OH concentration to 0. The length of the daytime period is calculated from the day of 360 year and location. Atmospheric ionization rates due to galactic cosmic rays are calculated as a function of latitude, altitude, and solar cycle phase by a model of energetic particle transport in the Earth's atmosphere (O'Brien, 2005). The transformation between geographic and geomagnetic coordinates is calculated with GEOPACK (http://geo.phys.spbu.ru/~tsyganenko/modeling.html) and the International Geomagnetic 365 Reference Field 12 coefficients (https://www.ngdc.noaa.gov/IAGA/vmod/igrf.html).

Aerosol nucleation rates calculated from the experimental thermochemical data of neutral and charged $H_2SO_4$/$H_2O$ cluster formation that are used in MAIA (Kazil and Lovejoy, 2007) compare well with neutral and charged $H_2SO_4$/$H_2O$ nucleation rates measured in the European

Organization for Nuclear Research (CERN) Cosmics Leaving Outdoor Droplets (CLOUD) chamber (Kirkby et al., 2011). Global model simulations, either using a parameterization of neutral and charged $H_2SO_4/H_2O$ nucleation based on the CERN CLOUD chamber measurements, or nucleation rates calculated from the experimental thermochemical data used in MAIA (Kazil et al., 2010) show a good agreement in the global mean profile of total (> 3 nm)
aerosol concentration (Määttänen et al., 2018).

MAIA is run along back-trajectories, initiated at the aircraft location, which were calculated using the Traj3D trajectory model (Bowman, 1993)and the National Center for Environmental Prediction (NCEP) global forecast system (GFS) meteorology (2015). NCEP provides
temperature, relative humidity, and pressure along the trajectories for the MAIA runs. The initial $SO_2$ concentration and the $H_2SO_4$ condensation sink of the initial aerosol size distribution were estimated from ATom observations at similar latitudes and altitudes. The geometric mean diameter (46 nm) and geometric standard deviation (2.8) of the initial aerosol size distribution were obtained by fitting a lognormal mode to the size distribution observed at the ATom
measurement locations. The noon concentration of OH in the simulations was set to $3 \times 10^6$ molec. cm$^{-3}$. This estimate agrees well with aircraft-measured OH concentrations during ATom (Kupc et al., 2020)."

*The comparison between CEDS emission rates and observed aerosol concentrations in Figure 11 is problematic since it in now way takes into account the transport of SO2. As mentioned in the text, observations were made mostly outside of the flight corridors and there the number concentrations of ultrafine particles did not correlate with SO2 concentration. Can it be that new particle formation occurred near flight corridors and these particles were transported to the regions of aircraft observations?*

Regarding Fig 11 and the comparison of measured SO2 flux with CEDS emissions, we have indeed omitted to explicitly address transport. We have noted in the manuscript the lifetime of $SO_2$ as around 1 month (line 394, and SM section 2) and zonal transport from flight corridors to the most distant observations from those corridors is on the order of half a day (line 386). Using the median observed windspeed of 10-30 m/s (line385) we can also see that the complete zonal
mixing timescale is on the order for 1 month, approximately equal to the lifetime of SO2. Since the jet core with peak wind speeds of 40-60 m/s is typically located on the tropopause in the 30-50N region (Manney et al., 2014), and the peak in SO2 is also located in this region, the zonal mixing time could be as little as one to two weeks. We believe this justifies the presentation of zonally averaged $SO_2$ in Fig 11 parts b and c. We agree with the referee that this was not
adequately addressed in the original manuscript and so propose to include the following explanation at line 386:
"Based on these measured windspeeds, zonal mixing is expected on timescales of around 1 month, which is approximately equal to the lifetime of $SO_2$. However, since the jet core with peak wind speeds of 40-60 m/s is typically located on the tropopause in the 30-50N region
(Manney et al., 2014), and the peak in SO2 is also located in this region, the zonal mixing time could be as little as one to two weeks. For this reason, we present zonally averaged SO2 concentrations in figure 11b and c."

The referee asks an interesting question about whether NPF could be occurring within flight
corridors and then transported to the ATom flight paths where we observed them. It is important to note that SO2 lifetimes are estimated to be much larger than nucleation mode aerosol lifetimes (1 month (line 394) for SO2, compared with a few days (line 147)) for nucleation mode aerosols). We therefore do not see a mechanism for transporting newly formed particles from regions of higher SO2 concentrations to regions of lower SO2 concentrations, given the assumption that SO2 is driving nucleation. Where there is a lack of correlation between SO2 and nucleation mode aerosol, we believe this is more likely to be due a different chemical source for aerosol nucleation, lack of measurement precision or the different sinks and lifetimes of SO2 and aerosols.

*Page 2, Lines 36-37: Solomon et al., 2011 does not discuss aerosol size distributions. Wouldn't Williamson et al., 2019 reference be more suitable reference here?*
We thank the referee for catching our mistake with the Solomon et al 2011 on lines 36-37. We think the Williamson et al 2019 reference is not ideal here, because that paper concerns ATom observations in the tropics, which generally did not penetrate into the stratosphere. Instead, we
have corrected this to:
"Aerosols in the lowermost stratosphere (LMS) are highly variable, even in the absence of major volcanic eruptions (Solomon et al., 2011), and models currently struggle to reproduce observed aerosol size distributions in this region of the atmosphere (Murphy et al., 2020)."

*Page 2, Lines 126-130: Where is this information about condensation and coagulation rates used?*
The condensation rates explained in the methods section line 126-130 are used to calculate the condensation sink for comparison with the MAIA box modeling presented in Figure 7 and discussed at lines 170 and 203-230. The coagulation rates are used in the supplementary material section S2 to estimate the lifetime of particles in the LMS (Fig S4), which is then referenced in
the main manuscript at line 147 and subsequently. To clarify this in the text we have added the following sentence at line 132:
"Condensation and coagulation rates will be used in this analysis to relate our observations to theory and models, and to estimate particle lifetimes."

*Page 5, Line 146: oder → order*
We have corrected the spelling mistake on line 146 and thank the referee for spotting that.

*Page 8, Lines 229-237: Here you discuss that the SO2 concentrations (40 pptv) were higher than the median observed values. Why was this value chosen? Was this level required to initiate nucleation? It is said that SO2*
*concentrations decreased to the observed concentrations. Was NPF still ongoing at these levels?*
MAIA was run only at a few discrete levels SO2 because of constraints on computing time. The pertinent levels were 20 and 40 pptv, results of which are shown in Fig 7. As the referee points out, this is above the median observed values. NPF did occur at 20 pptv, but slower growth meant that, for higher condensation sinks, observable increases in number concentrations of
particles larger than 2.65 nm were only seen on trajectories with higher RHw at these SO2 concentrations (Fig. 7f). At lower condensation sinks, increases in number concentrations of particles larger than 2.65 nm were clearly observable with SO2 at 20 pptv (Fig. 7h).

*Section 6: The first three paragraphs explain how aerosol emissions are calculated. The motivation for this*
*procedure is unclear to me. Is this done in order to obtain higher temporal resolution emission rates from CEDS monthly fields?*
In section 6 the first paragraph explains why we are considering aircraft as sources for small particles and SO2. The second paragraph introduces the CEDS database and explains the relevant sources used therein. The third paragraph details how we calculate expected emissions of nucleation mode aerosol using the CEDS database, using the data on aircraft emissions therein, and literature values to convert from SO2 emissions to nucleation mode particle emissions, since the nucleation mode particle emissions were not, to our knowledge, included in the CEDS database. We would like to understand better the aspects of this that are unclear, or potentially duplicating CEDS fields in order to make any necessary corrections.

*Page 13, Line 400: Can higher observed SO2 concentrations be due to transport from regions with higher emissions?*
In the SH LMS, the higher observed SO2 concentrations relative to CEDS emissions referred to on line 400 may well be due to transport, or perhaps small, more local sources, but we do not
have the data to make any firm conclusions regarding this.

*Page 14, Line 430: "influence from fires may suppress NPF in the LMS, or that the additional surface area from biomass burning particles shortens the lifetime of newly formed particles". Aren't these two the same thing?*
On line 430, we were attempting to reference the influence of biomass burning on two distinct
processes – the nucleation of aerosols, and the subsequent growth of those particles. The referee has shown that this lacked clarity, so we have modified the sentence from the original "suggesting that the influence from fires may suppress NPF in the LMS, or that the additional surface area from biomass burning particles shortens the lifetime of newly formed particles" to
"suggesting that the additional surface area from biomass burning particles may reduce nucleation mode number concentrations in the LMS through two mechanisms, suppressing the formation of particles by increasing the condensation sink, and shortening the lifetime of particles that do form by increasing the coagulation sink."

**Unsolicited corrections**

We would like to alert the referee to some unsolicited corrections we would like to make to the manuscript that came up in the course of addressing referee comments. These are detailed here below. We would also like to alert the referee to proposed corrections in our response to referee
#1.

Extraneous comma removed line 57

For clearer reading, Line 62 change from
"Ammonia and amines have been shown to contribute to NPF"
to
"Ammonia and amines have been shown to be involved in NPF"

Line 82: "lowermost stratosphere" changed to "LMS" for consistency.

Line 123 "CH3Cl" corrected to "$CH_3Cl$"

Line 127
"using Fuchs expression for the coagulation rate coefficient(Seinfeld and Pandis, 2006)"
corrected to
"using the Fuchs expression for the coagulation rate coefficient (Seinfeld and Pandis, 2006)"

Line 141 changed from "in' to "by" in "This stratospheric definition is consistent with that used by Murphy et al. (2020)".

We noted a missing parenthesis on line 195 and so added this.

Line 227, alteration made in order to read better from
"20 of the 55 trajectories experience more humid air, indicative of UT or tropopause conditions."
to
"The other 20 trajectories experienced more humid air, indicative of UT or tropopause conditions."

The full description of Asian Summer Monsoon (ASM) has been placed on the first usage at line 253 instead of where it was mistakenly put on line 424.

Line 309 "Because the lifetime of these particles is ~ days" changed to "Because the lifetime of these particles is on the order of days"

Line 341 "from" added to "through NPF resulting from $SO_2$ oxidation"

Line 402 "higher observed concentrations … are also likely" corrected from "is also likely"

Line 405 "we get a total flux" change to "we determine a total flux"

Line 435 "ASM-sources particles" corrected to "ASM-sourced particles"
Line 465 "volcano emissions" changed to "volcanic emissions"

Line 451 "the only eruption to occur between the tropopause height: corrected to "the only eruption to reach between the tropopause height"

For clarity, line 495 has been rewritten from
"How this would be achieved, and the potential consequences, both intended effects and side-effects are highly uncertain."
to
"How this would be achieved, and the potential consequences of such actions (both the intended effects and any unintended side-effects) are highly uncertain."

We noted in the SM line 44 a forward slash had accidentally been used instead of a period, and have corrected this.

Additional references have been added at line 53 and the order of references changed to

"New particle formation (NPF) has been well documented in a variety of locations in the planetary boundary layer and free troposphere (Clarke et al., 1998;Clarke et al., 2013;Kulmala et al., 2013;Williamson et al., 2019)."

Similarly, at line 284 which now reads
New particles have previously been shown to … form in the tropical UT (Clarke et al., 1998;Clarke and Kapustin, 2002;Clarke et al., 2013;Williamson et al., 2019)

Figure 3: "level" removed from a,b y-axis and corrections made to caption text: "except for" changed to "except", "grew" changed to "grey"

Figure 4: y-axes for NH and SH changed to be the same range to make comparison easier

Figure 8: greater than and less than symbols in legend corrected

Figure S3. We have added a more through explanation of the figure and how this was calculated from the data in the caption, and included an example mass spectrum to illustrate this. The new proposed figure and caption are as follows:

[Figure]

Figure S2: Aerosol acidity in the LMS.   a) An example negative ion spectrum of an acidic sulphate particle. This spectrum is from a 0.39 μm diameter particle in the stratosphere at 12.2 km and 310 ppbv of $O_3$ on 20171009. Laboratory calibrations show that the $H_2SO_4 \bullet HSO_4^-$ peak is very small or non-existent for particles composed of ammonium sulphate and the cluster ion peak increases with acidity until it is a large peak for nearly pure sulfuric acid. b) The bars for the NH (left) and SH (right), separated by season, show the average ratio of the size of the cluster peak at m/z 195 to the sum of the peaks at m/z 195 and 97. The averages are for particles when $O_3$ concentrations were 250 to 350 ppbv in the stratosphere. The averages are also for particles between 0.35 and 0.6 μm diameter because in the stratosphere most particles of that size originated in the stratosphere. Lab calibrations of particles composed of $(NH_4)_{0.25}H_{1.75}SO_4$ had a negative ion ration m/z 195/(97+195) of 0.034, therefore we consider ratios higher than this (more acidic) to contain less than 0.25 mole fraction ammonium. Rather than analyzing possible differences in the acidity with season, here we emphasize that stratospheric particles in all seasons and both hemispheres are highly acidic. This sets limits on the possible concentration of gas phase ammonia. Calculations of uptake from the gas phase show that a continuous 1 pptv of gas phase ammonia could add 0.25 mol fraction ammonium to sulfuric acid particles in less than a week.

**Response to Editor**

Thank you for recommending the clarification of the latitudinal extent of this study. We have accordingly changed the title from "Large hemispheric difference in ultrafine aerosol concentrations in the lowermost stratosphere" to "Large hemispheric difference in ultrafine aerosol concentrations in the lowermost stratosphere at mid and high latitudes"
in both the main manuscript and supplementary material. In the main manuscript we have also changed line 18 in the abstract to read "we identify a mode of ultrafine aerosol in the lowermost stratosphere (LMS) at mid and high latitudes", and lines 469 and 470 in the conclusion to read "

the most likely cause of elevated numbers of ultrafine aerosol in the NH LMS at mid and high latitudes is aviation".

We also corrected a mistake in the author contributions section, and so changed line 516 from "GD (water vapour)" to "GSD and JPD (water vapour)".

**References**

NCEP GFS 0.25-degree global forecast grids historical archive. Systems, N. U. R. D. A. C. a. I., and Lab (Eds.), 2015.

Bowman, K. P.: Large-Scale Isentropic Mixing Properties of the Antarctic Polar Vortex from Analyzed Winds, J Geophys Res-Atmos, 98, 23013-23027, Doi 10.1029/93jd02599, 1993.

Clarke, A. D., Varner, J. L., Eisele, F., Mauldin, R. L., Tanner, D., and Litchy, M.: Particle
production in the remote marine atmosphere: Cloud outflow and subsidence during ACE 1, J Geophys Res-Atmos, 103, 16397-16409, Doi 10.1029/97jd02987, 1998.

Clarke, A. D., and Kapustin, V. N.: A pacific aerosol survey. Part I: A decade of data on particle production, transport, evolution, and mixing in the troposphere, Journal of the Atmospheric Sciences, 59, 363-382, Doi 10.1175/1520-0469(2002)059<0363:Apaspi>2.0.Co;2, 2002.

Clarke, A. D., Freitag, S., Simpson, R. M. C., Hudson, J. G., Howell, S. G., Brekhovskikh, V. L., Campos, T., Kapustin, V. N., and Zhou, J.: Free troposphere as a major source of CCN for the equatorial pacific boundary layer: long-range transport and teleconnections, Atmos Chem Phys, 13, 7511-7529, 10.5194/acp-13-7511-2013, 2013.

Clegg, S. L., Rard, J. A., and Pitzer, K. S.: Thermodynamic properties of 0–6 mol kg–1 aqueous sulfuric acid from 273.15 to 328.15 K, J. Chem. Soc., Faraday Trans., 90, 1875-1894, 10.1039/FT9949001875, 1994.

Curtius, J., Froyd, K. D., and Lovejoy, E. R.: Cluster Ion Thermal Decomposition (I): Experimental Kinetics Study and ab Initio Calculations for HSO4-(H2SO4)x(HNO3)y, The Journal of Physical Chemistry A, 105, 10867-10873, 10.1021/jp0124950, 2001.

Froyd, K. D., and Lovejoy, E. R.: Experimental Thermodynamics of Cluster Ions Composed of H2SO4 and H2O. 1. Positive Ions, The Journal of Physical Chemistry A, 107, 9800-9811, 10.1021/jp027803o, 2003.

Giauque, W. F., Hornung, E. W., Kunzler, J. E., and Rubin, T. R.: The Thermodynamic Properties of Aqueous Sulfuric Acid Solutions and Hydrates from 15 to 300°K.1, Journal of the American 635 Chemical Society, 82, 62-70, 10.1021/ja01486a014, 1960.

Hanson, D. R., and Lovejoy, E. R.: Measurement of the Thermodynamics of the Hydrated Dimer and Trimer of Sulfuric Acid, The Journal of Physical Chemistry A, 110, 9525-9528, 10.1021/jp062844w, 2006.

Kazil, J., and Lovejoy, E. R.: A semi-analytical method for calculating rates of new sulfate aerosol 640 formation from the gas phase, Atmos. Chem. Phys., 7, 3447-3459, 10.5194/acp-7-3447-2007, 2007.

Kazil, J., Lovejoy, E. R., Jensen, E. J., and Hanson, D. R.: Is aerosol formation in cirrus clouds possible?, Atmos. Chem. Phys., 7, 1407-1413, 10.5194/acp-7-1407-2007, 2007.

Kazil, J., Stier, P., Zhang, K., Quaas, J., Kinne, S., O'Donnell, D., Rast, S., Esch, M., Ferrachat, S., 645 Lohmann, U., and Feichter, J.: Aerosol nucleation and its role for clouds and Earth's radiative forcing in the aerosol-climate model ECHAM5-HAM, Atmos Chem Phys, 10, 10733-10752, 10.5194/acp-10-10733-2010, 2010.

Kirkby, J., Curtius, J., Almeida, J., Dunne, E., Duplissy, J., Ehrhart, S., Franchin, A., Gagné, S., Ickes, L., Kürten, A., Kupc, A., Metzger, A., Riccobono, F., Rondo, L., Schobesberger, S., 650 Tsagkogeorgas, G., Wimmer, D., Amorim, A., Bianchi, F., Breitenlechner, M., David, A., Dommen, J., Downard, A., Ehn, M., Flagan, R. C., Haider, S., Hansel, A., Hauser, D., Jud, W., Junninen, H., Kreissl, F., Kvashin, A., Laaksonen, A., Lehtipalo, K., Lima, J., Lovejoy, E. R., Makhmutov, V., Mathot, S., Mikkilä, J., Minginette, P., Mogo, S., Nieminen, T., Onnela, A., Pereira, P., Petäjä, T., Schnitzhofer, R., Seinfeld, J. H., Sipilä, M., Stozhkov, Y., Stratmann, F., 655 Tomé, A., Vanhanen, J., Viisanen, Y., Vrtala, A., Wagner, P. E., Walther, H., Weingartner, E., Wex, H., Winkler, P. M., Carslaw, K. S., Worsnop, D. R., Baltensperger, U., and Kulmala, M.: Role of sulphuric acid, ammonia and galactic cosmic rays in atmospheric aerosol nucleation, Nature, 476, 429-433, 10.1038/nature10343, 2011.

Kulmala, M., Kontkanen, J., Junninen, H., Lehtipalo, K., Manninen, H. E., Nieminen, T., Petaja, T., Sipila, M., Schobesberger, S., Rantala, P., Franchin, A., Jokinen, T., Jarvinen, E., Aijala, M., Kangasluoma, J., Hakala, J., Aalto, P. P., Paasonen, P., Mikkila, J., Vanhanen, J., Aalto, J., Hakola, H., Makkonen, U., Ruuskanen, T., Mauldin, R. L., Duplissy, J., Vehkamaki, H., Back, J., Kortelainen, A., Riipinen, I., Kurten, T., Johnston, M. V., Smith, J. N., Ehn, M., Mentel, T. F., Lehtinen, K. E. J., Laaksonen, A., Kerminen, V. M., and Worsnop, D. R.: Direct Observations of
Atmospheric Aerosol Nucleation, Science, 339, 943-946, 10.1126/science.1227385, 2013.

Kupc, A., Williamson, C. J., Hodshire, A. L., Kazil, J., Ray, E., Bui, T. P., Dollner, M., Froyd, K. D., McKain, K., Rollins, A., Schill, G. P., Thames, A., Weinzierl, B. B., Pierce, J. R., and Brock, C. A.: The potential role of organics in new particle formation and initial growth in the remote tropical upper troposphere, Atmos. Chem. Phys., 20, 15037-15060, 10.5194/acp-20-15037-2020, 2020.

Lovejoy, E. R., Hanson, D. R., and Huey, L. G.: Kinetics and Products of the Gas-Phase Reaction of $SO_3$ with Water, The Journal of Physical Chemistry, 100, 19911-19916, 10.1021/jp962414d, 1996.

Lovejoy, E. R., and Curtius, J.: Cluster Ion Thermal Decomposition (II):  Master Equation Modeling in the Low-Pressure Limit and Fall-Off Regions. Bond Energies for $HSO_4-$
$(H_2SO_4)_x(HNO_3)_y$, The Journal of Physical Chemistry A, 105, 10874-10883, 10.1021/jp012496s, 2001.

Lovejoy, E. R., Curtius, J., and Froyd, K. D.: Atmospheric ion-induced nucleation of sulfuric acid and water, J Geophys Res-Atmos, 109, Artn D08204 10.1029/2003jd004460, 2004.

Määttänen, A., Merikanto, J., Henschel, H., Duplissy, J., Makkonen, R., Ortega, I. K., and Vehkamäki, H.: New Parameterizations for Neutral and Ion-Induced Sulfuric Acid-Water Particle Formation in Nucleation and Kinetic Regimes, Journal of Geophysical Research: Atmospheres, 123, 1269-1296, https://doi.org/10.1002/2017JD027429, 2018.

Manney, G. L., Hegglin, M. I., Daffer, W. H., Schwartz, M. J., Santee, M. L., and Pawson, S.:
Climatology of Upper Tropospheric–Lower Stratospheric (UTLS) Jets and Tropopauses in MERRA, J Climate, 27, 3248-3271, 10.1175/jcli-d-13-00243.1, 2014.

Murphy, D. M., Froyd, K. D., Bourgeois, I., Brock, C. A., Kupc, A., Peischl, J., Schill, G. P., Thompson, C. R., Williamson, C. J., and Yu, P.: Radiative and chemical implications of the size and composition of aerosol particles in the existing or modified global stratosphere, Atmos.
Chem. Phys. Discuss., 2020, 1-32, 10.5194/acp-2020-909, 2020.

O'Brien, K.: The theory of cosmic-ray and high-energy solar-particle transport in the atmosphere, in, 29-44, 2005.

Seinfeld, J. H., and Pandis, S. N.: Atmospheric chemistry and physics from air pollution to climate change, 2nd ed., Wiley, Hoboken, N.J., 1203 S. pp., 2006.

Solomon, S., Daniel, J. S., Neely, R. R., Vernier, J.-P., Dutton, E. G., and Thomason, L. W.: The Persistently Variable "Background" Stratospheric Aerosol Layer and Global Climate Change, Science, 333, 866-870, 10.1126/science.1206027, 2011.

Williamson, C. J., Kupc, A., Axisa, D., Bilsback, K. R., Bui, T., Campuzano-Jost, P., Dollner, M., Froyd, K. D., Hodshire, A. L., Jimenez, J. L., Kodros, J. K., Luo, G., Murphy, D. M., Nault, B. A., Ray,
E. A., Weinzierl, B., Wilson, J. C., Yu, F. Q., Yu, P. F., Pierce, J. R., and Brock, C. A.: A large source of cloud condensation nuclei from new particle formation in the tropics, Nature, 574, 399-+, 10.1038/s41586-019-1638-9, 2019.

**References**

Brock, C. A., Schroder, F., Karcher, B., Petzold, A., Busen, R., and Fiebig, M.: Ultrafine particle
size distributions measured in aircraft exhaust plumes, J Geophys Res-Atmos, 105, 26555-26567, Doi 10.1029/2000jd900360, 2000.

Brock, C. A., Williamson, C., Kupc, A., Froyd, K. D., Erdesz, F., Wagner, N., Richardson, M., Schwarz, J. P., Gao, R. S., Katich, J. M., Campuzano-Jost, P., Nault, B. A., Schroder, J. C., Jimenez, J. L., Weinzierl, B., Dollner, M., Bui, T., and Murphy, D. M.: Aerosol size distributions during the
Atmospheric Tomography Mission (ATom): methods, uncertainties, and data products, Atmos Meas Tech, 12, 3081-3099, 10.5194/amt-12-3081-2019, 2019.

Clarke, A., and Kapustin, V.: Hemispheric aerosol vertical profiles: Anthropogenic impacts on optical depth and cloud nuclei (vol 329, pg 1488, 2010), Science, 330, 1047-1047, 2010.

Clarke, A. D., Varner, J. L., Eisele, F., Mauldin, R. L., Tanner, D., and Litchy, M.: Particle
production in the remote marine atmosphere: Cloud outflow and subsidence during ACE 1, J Geophys Res-Atmos, 103, 16397-16409, Doi 10.1029/97jd02987, 1998.

Clarke, A. D., Eisele, F., Kapustin, V. N., Moore, K., Tanner, D., Mauldin, L., Litchy, M., Lienert, B., Carroll, M. A., and Albercook, G.: Nucleation in the equatorial free troposphere: Favorable environments during PEM-Tropics, J Geophys Res-Atmos, 104, 5735-5744, Doi
10.1029/98jd02303, 1999.

Clarke, A. D., and Kapustin, V. N.: A pacific aerosol survey. Part I: A decade of data on particle production, transport, evolution, and mixing in the troposphere, Journal of the Atmospheric Sciences, 59, 363-382, Doi 10.1175/1520-0469(2002)059<0363:Apaspi>2.0.Co;2, 2002.

Clarke, A. D., Freitag, S., Simpson, R. M. C., Hudson, J. G., Howell, S. G., Brekhovskikh, V. L.,
Campos, T., Kapustin, V. N., and Zhou, J.: Free troposphere as a major source of CCN for the equatorial pacific boundary layer: long-range transport and teleconnections, Atmos Chem Phys, 13, 7511-7529, 10.5194/acp-13-7511-2013, 2013.

Curtius, J., Froyd, K. D., and Lovejoy, E. R.: Cluster Ion Thermal Decomposition (I): Experimental Kinetics Study and ab Initio Calculations for HSO4-(H2SO4)x(HNO3)y, The Journal of Physical Chemistry A, 105, 10867-10873, 10.1021/jp0124950, 2001.

Froyd, K. D., and Lovejoy, E. R.: Experimental Thermodynamics of Cluster Ions Composed of H2SO4 and H2O. 1. Positive Ions, The Journal of Physical Chemistry A, 107, 9800-9811, 10.1021/jp027803o, 2003.

Hanson, D. R., and Lovejoy, E. R.: Measurement of the Thermodynamics of the Hydrated Dimer and Trimer of Sulfuric Acid, The Journal of Physical Chemistry A, 110, 9525-9528, 10.1021/jp062844w, 2006.

Kazil, J., and Lovejoy, E. R.: A semi-analytical method for calculating rates of new sulfate aerosol formation from the gas phase, Atmos. Chem. Phys., 7, 3447-3459, 10.5194/acp-7-3447-2007, 2007.

Kazil, J., Stier, P., Zhang, K., Quaas, J., Kinne, S., O'Donnell, D., Rast, S., Esch, M., Ferrachat, S., Lohmann, U., and Feichter, J.: Aerosol nucleation and its role for clouds and Earth's radiative forcing in the aerosol-climate model ECHAM5-HAM, Atmos Chem Phys, 10, 10733-10752, 10.5194/acp-10-10733-2010, 2010.

Kirkby, J., Curtius, J., Almeida, J., Dunne, E., Duplissy, J., Ehrhart, S., Franchin, A., Gagné, S., Ickes, L., Kürten, A., Kupc, A., Metzger, A., Riccobono, F., Rondo, L., Schobesberger, S., Tsagkogeorgas, G., Wimmer, D., Amorim, A., Bianchi, F., Breitenlechner, M., David, A., Dommen, J., Downard, A., Ehn, M., Flagan, R. C., Haider, S., Hansel, A., Hauser, D., Jud, W., Junninen, H., Kreissl, F., Kvashin, A., Laaksonen, A., Lehtipalo, K., Lima, J., Lovejoy, E. R., Makhmutov, V., Mathot, S., Mikkilä, J., Minginette, P., Mogo, S., Nieminen, T., Onnela, A., Pereira, P., Petäjä, T., Schnitzhofer, R., Seinfeld, J. H., Sipilä, M., Stozhkov, Y., Stratmann, F., Tomé, A., Vanhanen, J., Viisanen, Y., Vrtala, A., Wagner, P. E., Walther, H., Weingartner, E., Wex, H., Winkler, P. M., Carslaw, K. S., Worsnop, D. R., Baltensperger, U., and Kulmala, M.: Role of sulphuric acid, ammonia and galactic cosmic rays in atmospheric aerosol nucleation, Nature, 476, 429-433, 10.1038/nature10343, 2011.

Kulmala, M., Kontkanen, J., Junninen, H., Lehtipalo, K., Manninen, H. E., Nieminen, T., Petaja, T., Sipila, M., Schobesberger, S., Rantala, P., Franchin, A., Jokinen, T., Jarvinen, E., Aijala, M., Kangasluoma, J., Hakala, J., Aalto, P. P., Paasonen, P., Mikkila, J., Vanhanen, J., Aalto, J., Hakola, H., Makkonen, U., Ruuskanen, T., Mauldin, R. L., Duplissy, J., Vehkamaki, H., Back, J., Kortelainen, A., Riipinen, I., Kurten, T., Johnston, M. V., Smith, J. N., Ehn, M., Mentel, T. F., Lehtinen, K. E. J., Laaksonen, A., Kerminen, V. M., and Worsnop, D. R.: Direct Observations of Atmospheric Aerosol Nucleation, Science, 339, 943-946, 10.1126/science.1227385, 2013.

Kupc, A., Williamson, C. J., Hodshire, A. L., Kazil, J., Ray, E., Bui, T. P., Dollner, M., Froyd, K. D., McKain, K., Rollins, A., Schill, G. P., Thames, A., Weinzierl, B. B., Pierce, J. R., and Brock, C. A.: The potential role of organics in new particle formation and initial growth in the remote tropical upper troposphere, Atmos. Chem. Phys., 20, 15037-15060, 10.5194/acp-20-15037-2020, 2020.

Kupc, A., Williamson, C.J., Hodshire, A. L  Jan Kazil, Eric Ray, T. Paul Bui, Maximilian Dollner, Karl D. Froyd, Kathryn McKain, Andrew Rollins, Gregory P. Schill, Alexander Thames, Bernadett B. Weinzierl, Jeffrey R. Pierce and Charles A. Brock: The potential role of organics in new particle formation and initial growth in the remote tropical upper troposphere, submitted, 2020.

Lovejoy, E. R., and Curtius, J.: Cluster Ion Thermal Decomposition (II):  Master Equation Modeling in the Low-Pressure Limit and Fall-Off Regions. Bond Energies for HSO4-(H2SO4)x(HNO3)y, The Journal of Physical Chemistry A, 105, 10874-10883, 10.1021/jp012496s, 2001.

Määttänen, A., Merikanto, J., Henschel, H., Duplissy, J., Makkonen, R., Ortega, I. K., and
Vehkamäki, H.: New Parameterizations for Neutral and Ion-Induced Sulfuric Acid-Water Particle Formation in Nucleation and Kinetic Regimes, Journal of Geophysical Research: Atmospheres, 123, 1269-1296, https://doi.org/10.1002/2017JD027429, 2018.

Murphy, D. M., Froyd, K. D., Bourgeois, I., Brock, C. A., Kupc, A., Peischl, J., Schill, G. P., Thompson, C. R., Williamson, C. J., and Yu, P.: Radiative and chemical implications of the size
and composition of aerosol particles in the existing or modified global stratosphere, Atmos. Chem. Phys. Discuss., 2020, 1-32, 10.5194/acp-2020-909, 2020.

Rollins, A. W., Thornberry, T. D., Watts, L. A., Yu, P., Rosenlof, K. H., Mills, M., Baumann, E., Giorgetta, F. R., Bui, T. V., Höpfner, M., Walker, K. A., Boone, C., Bernath, P. F., Colarco, P. R., Newman, P. A., Fahey, D. W., and Gao, R. S.: The role of sulfur dioxide in stratospheric aerosol
formation evaluated by using in situ measurements in the tropical lower stratosphere, Geophys. Res. Lett., 44, 4280-4286, 10.1002/2017gl072754, 2017.

Schroder, F., Brock, C. A., Baumann, R., Petzold, A., Busen, R., Schulte, P., and Fiebig, M.: In situ studies on volatile jet exhaust particle emissions: Impact of fuel sulfur content and environmental conditions on nuclei mode aerosols, J Geophys Res-Atmos, 105, 19941-19954,
Doi 10.1029/2000jd900112, 2000.

Seinfeld, J. H., and Pandis, S. N.: Atmospheric chemistry and physics from air pollution to climate change, 2nd ed., Wiley, Hoboken, N.J., 1203 S. pp., 2006.

Williamson, C. J., Kupc, A., Axisa, D., Bilsback, K. R., Bui, T., Campuzano-Jost, P., Dollner, M., Froyd, K. D., Hodshire, A. L., Jimenez, J. L., Kodros, J. K., Luo, G., Murphy, D. M., Nault, B. A., Ray,
E. A., Weinzierl, B., Wilson, J. C., Yu, F. Q., Yu, P. F., Pierce, J. R., and Brock, C. A.: A large source of cloud condensation nuclei from new particle formation in the tropics, Nature, 574, 399-+, 10.1038/s41586-019-1638-9, 2019.

Wofsy, S. C., Afshar, S., Allen, H. M., Apel, E., Asher, E. C., Barletta, B., Bent, J., Bian, H., Biggs, B. C., Blake, D. R., Blake, N., Bourgeois, I., Brock, C. A., Brune, W. H., Budney, J. W., Bui, T. P.,
Butler, A., Campuzano-Jost, P., Chang, C. S., Chin, M., Commane, R., Correa, G., Crounse, J. D., Cullis, P. D., Daube, B. C., Day, D. A., Dean-Day, J. M., Dibb, J. E., Digangi, J. P., Diskin, G. S., Dollner, M., Elkins, J. W., Erdesz, F., Fiore, A. M., Flynn, C. M., Froyd, K., Gesler, D. W., Hall, S. R.,

Hanisco, T. F., Hannun, R. A., Hills, A. J., Hintsa, E. J., Hoffman, A., Hornbrook, R. S., Huey, L. G., Hughes, S., Jimenez, J. L., Johnson, B. J., Katich, J. M., Keeling, R., Kim, M. J., Kupc, A., Lait, L. R.,

Lamarque, J. F., Liu, J., McKain, K., McLaughlin, R. J., Meinardi, S., Miller, D. O., Montzka, S. A., Moore, F. L., Morgan, E. J., Murphy, D. M., Murray, L. T., Nault, B. A., Neuman, J. A., Newman, P. A., Nicely, J. M., Pan, X., Paplawsky, W., Peischl, J., Prather, M. J., Price, D. J., Ray, E., Reeves, J. M., Richardson, M., Rollins, A. W., Rosenlof, K. H., Ryerson, T. B., Scheuer, E., Schill, G. P., Schroder, J. C., Schwarz, J. P., St.Clair, J. M., Steenrod, S. D., Stephens, B. B., Strode, S. A.,

Sweeney, C., Tanner, D., Teng, A. P., Thames, A. B., Thompson, C. R., Ullmann, K., Veres, P. R., Vizenor, N., Wagner, N. L., Watt, A., Weber, R., Weinzierl, B., Wennberg, P., Williamson, C. J., Wilson, J. C., Wolfe, G. M., Woods, C. T., and Zeng, L. H.: ATom: Merged Atmospheric Chemistry, Trace Gases, and Aerosols. ORNL Distributed Active Archive Center, 2018.